# AMPK and vacuole-associated Atg14p orchestrate µ-lipophagy for energy production and long-term survival under glucose starvation

Arnold Y Seo[1,2], Pick-Wei Lau[2], Daniel Feliciano[1,2], Prabuddha Sengupta[1,2], Mark A Le Gros[3,4], Bertrand Cinquin[3], Carolyn A Larabell[3,4], Jennifer Lippincott-Schwartz[1,2]*

[1]Janelia Research Campus, Howard Hughes Medical Institute, Ashburn, United States; [2]Cell Biology and Metabolism Program, National Institutes of Health, Bethesda, United States; [3]Department of Anatomy, University of California, San Francisco, San Francisco, United States; [4]Physical Biosciences Division, Lawrence Berkeley National Laboratory, Berkeley, United States

**Abstract** Dietary restriction increases the longevity of many organisms, but the cell signaling and organellar mechanisms underlying this capability are unclear. We demonstrate that to permit long-term survival in response to sudden glucose depletion, yeast cells activate lipid-droplet (LD) consumption through micro-lipophagy (µ-lipophagy), in which fat is metabolized as an alternative energy source. AMP-activated protein kinase (AMPK) activation triggered this pathway, which required Atg14p. More gradual glucose starvation, amino acid deprivation or rapamycin did not trigger µ-lipophagy and failed to provide the needed substitute energy source for long-term survival. During acute glucose restriction, activated AMPK was stabilized from degradation and interacted with Atg14p. This prompted Atg14p redistribution from ER exit sites onto liquid-ordered vacuole membrane domains, initiating µ-lipophagy. Our findings that activated AMPK and Atg14p are required to orchestrate µ-lipophagy for energy production in starved cells is relevant for studies on aging and evolutionary survival strategies of different organisms.

*For correspondence: lippincottschwartzj@janelia.hhmi.org

## Introducton

Nutrient-depleted cells can only persist long-term by initiating a two-pronged survival response: they need to be able to recycle cytoplasmic components and they need a source of energy available from within the cell. The first is achieved through self-digestion involving bulk autophagy pathways (*Green et al., 2011*; *Yen and Klionsky, 2008*), and the second requires a shift to metabolizing lipids for ATP production given the unavailability of glucose (*Hardie and Carling, 1997*; *Zechner et al., 2012*). Unless both responses are triggered, cell and organismal survival are jeopardized.

How bulk autophagy and lipid consumption pathways are coordinated to permit long-term survival in starved cells represents a fundamental and challenging question. Its answer is particularly relevant to aging research as caloric restriction and enhanced lifespan in various organisms are related (*Cantó and Auwerx, 2011*). But not all caloric restriction programs have the same beneficial effects on organismal health and survival (*Greer and Brunet, 2009*; *Wu et al., 2013*). For example, yeast cells starved by different nutrient regimes either die quickly or live long-term (*Goldberg et al., 2009*); mice strains with differing sensitivities to different nutrients show variable responses to caloric restriction (*Liao et al., 2010*); and non-human primates exhibit short or long lifespans on different

starvation diets (*Colman et al., 2014*). These results raise the possibility that cells mount distinct survival responses depending on the starvation condition, with only some regimens activating both lipid catabolism and autophagy, which are necessary for long-term survival.

In this study, we use imaging and functional analysis approaches to investigate the conditions, and signaling and organellar systems, that trigger bulk autophagy and lipid consumption pathways to permit long-term survival under starvation. Using the budding yeast *Saccharomyces cerevisiae* as our model system, we demonstrate that to initiate and sustain bulk autophagy and lipid droplet (LD) breakdown pathways together, starved cells need to sense an acute reduction in glucose levels. Cells undergoing gradual glucose reduction or mere amino acid starvation, by contrast, only induce bulk autophagy without initiating LD consumption, and do not survive long-term. We further show that LD consumption in cells undergoing acute glucose starvation occurs by the process of micro-autophagy of LDs (i.e. μ-lipophagy), which is dependent on AMPK activation and core autophagic machinery. Atg14p plays a particularly important role in this process. It shifts its distribution from ER exit sites (ERES) to liquid-ordered membrane domains on the vacuolar surface in response to AMPK activation where, together with Atg6p, it facilitates vacuole docking and internalization of LDs. Cells that cannot activate AMPK or that lack Atg14p or Atg6p do not deliver LDs into the vacuole for degradation and fail to thrive under acute glucose starvation. These findings highlight the importance of μ-lipophagy and its regulation for understanding the cellular mechanisms underlying lifespan extension under calorie restriction and show a fundamental plasticity in the regulation and function of core autophagy components in response to different metabolic or stress conditions.

## Results

### Cellular responses associated with prolonged lifespan under acute glucose restriction

Prior work in budding yeast has shown that different regimens of depleting glucose during starvation lead to dramatically different cellular lifespans (*Aris et al., 2013*; *Smith et al., 2007*). In particular, cells growing in synthetic minimal (SD) media (containing a restricted set of amino acids) with 2% glucose that are shifted into 0.4% glucose with no nutrient replenishment (i.e. acute glucose restriction, Acute GR) survive significantly longer than those placed in similar media containing 2% glucose (i.e. gradual glucose restriction, Gradual GR), even though a majority of nutrients become completely depleted within 1 day under both conditions. This surprising effect is shown in *Figure 1A*, with ~99.9% of cells starved by gradual GR dying within 9 days and nearly all cells starved by acute GR still alive after 25 days (*Figure 1A*). Thus, when starved of all nutrients, yeast cells survive differentially depending on whether they have sensed glucose being drained quickly or slowly out of the media.

To explore what metabolic features were associated with the increased survival of yeast cells undergoing rapid depletion of glucose from the media (i.e. Acute GR), we first examined changes in their cell respiration (i.e. oxygen consumption). Prior to glucose restriction, cells had very low oxygen consumption rates due to the cell's reliance on glycolysis (*Otterstedt et al., 2004*). By day 1 of GR, acutely-restricted cells had amplified their oxygen consumption rates to 2.2X that of gradually restricted cells (*Figure 1B*). Thereafter, acutely restricted cells maintained this higher respiratory rate, whereas gradually restricted cells exhibited a negligible oxygen consumption rate by day 3.

We also investigated the effect of oxidative stress on long-term cell viability under acute or gradual GR. Only cells undergoing acute GR remained viable after exposure to different oxidative stressors (i.e. hydrogen peroxide and menadione) (*Figure 1—figure supplement 1A*), with their viability resembling that of cells undergoing acute GR without stressors. Examining mitochondria morphology next, we observed that cells undergoing acute GR for 3 days had highly fused and elaborate mitochondria, whereas those undergoing gradual GR had exceedingly fragmented mitochondria (*Figure 1—figure supplement 1B and C*). Cells undergoing acute GR thus appear to have greater respiratory rates, increased stress resistance, and more highly fused mitochondria than cells undergoing gradual GR.

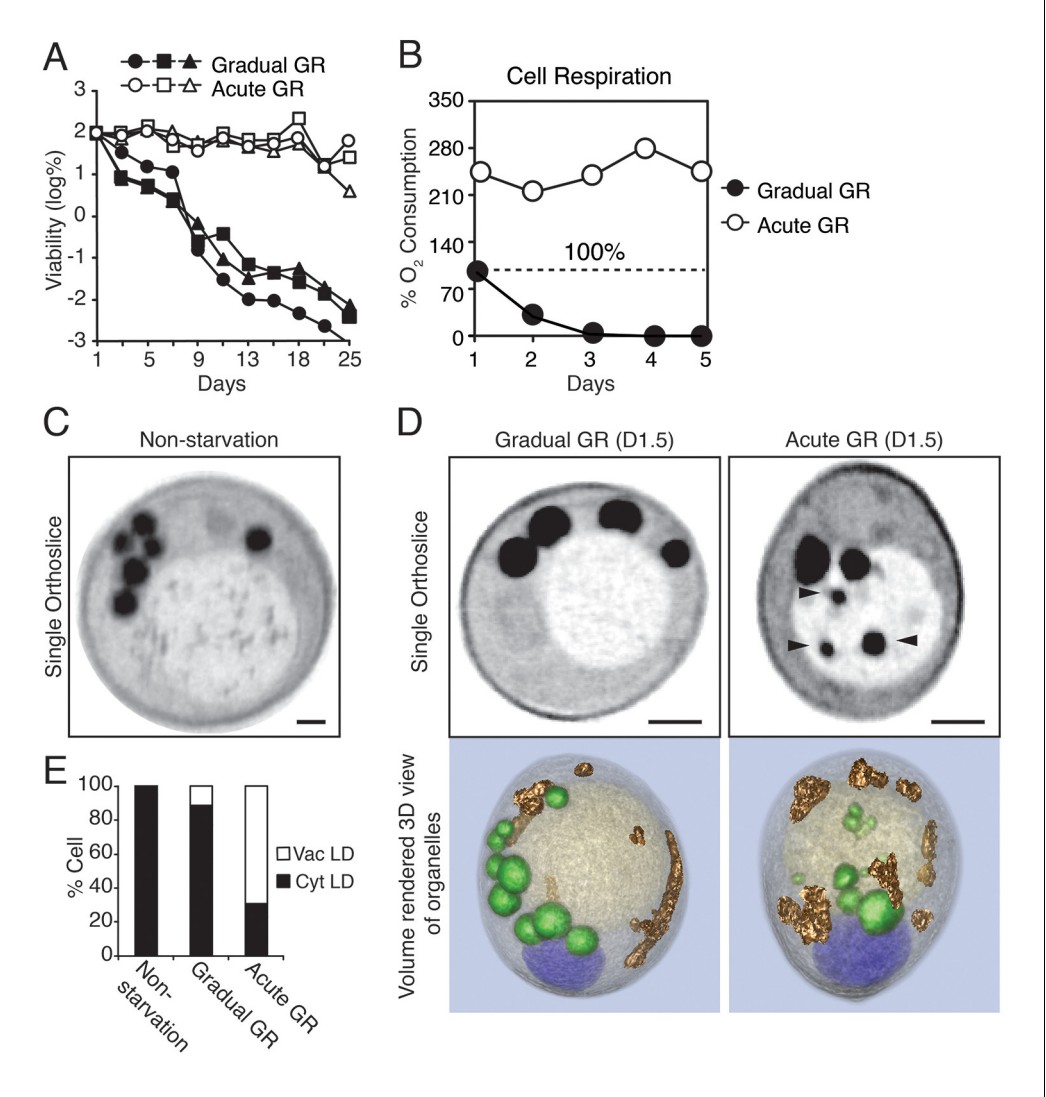

**Figure 1.** Starvation by acute GR increases cell survival and induces vacuolar LD delivery. (**A**) Long-term survival of cells undergoing gradual or acute GR was measured as described in the Materials and methods. Cell survival is plotted as the log of a percentage viable cell number at day 1 (which was set at 100%). Three biologically independent experiments are shown together. (**B**) Cell respiration was determined during a cell survival experiment described in the Materials and methods. $O_2$ consumption rate is plotted as a percentage of that seen in cells under gradual GR at day 1 (which was set at 100%). (**C**) Representative SXT orthoslice image of a yeast cell under non-starvation is shown. (**D**) Representative SXT orthoslice images of yeast cells under day 1.5 (D1.5) of gradual or acute GR are shown. Arrowheads indicate LDs inside the vacuole. Scale bar represents 0.5 μm. Lower panels show full 3D SXT images (LD: green; nucleus: purple; vacuole: pale yellow; mitochondria: gold). (**E**) Percentage of cells having only cytoplasmic LDs (Cyt LD) or having both Cyt LD and vacuole associated LDs (Vac LD) are shown. Data were analyzed from full 3D tomograms of the SXT images. Approximately 50 cells per each condition were analyzed.

The following figure supplements are available for figure 1:

**Figure supplement 1.** Starvation by acute GR enhances cellular oxidative stress resistance and induces mitochondrial tubulation.

**Figure supplement 2.** LDs and vacuolar lipase Atg15p are necessary for cell survival during starvation.

## Relationship between LDs and the vacuole under acute GR

A crucial way that cells alleviate metabolic demands during glucose deprivation is to form and consume LDs, whose fatty acid products fuel mitochondrial oxidative phosphorylation (OXPHOS) under starvation (*Singh et al., 2009*; *van Zutphen et al., 2014*). In studying this response, we found that cells genetically modified so they could not form LDs died quickly under glucose starvation (*Figure 1—figure supplement 2A*; also see [*Velázquez et al., 2016*]). Moreover, cells without vacuolar lipases had reduced survival rates under GR (*Figure 1—figure supplement 2B*; also see [*van Zutphen et al., 2014*]). These results suggested that metabolic reprogramming for long-term survival under glucose starvation requires the activities of LDs and the vacuole.

To visualize LDs and the vacuole under glucose starvation, we employed soft X-ray tomography (SXT). This technique gives a full 3D tomographic view of cryo-immobilized cells at 50 nm resolution, with no chemical fixation (*Larabell and Nugent, 2010*). Organelles are distinguishable based on the extent they absorb X-rays, with carbon-rich organelles (such as LDs) absorbing more X-rays and generating higher contrast than fluid-filled organelles, such as the yeast vacuole (*Uchida et al., 2011*).

In non-starved cells imaged using SXT, LDs (appearing as black spherical structures) were exclusively localized in the cytoplasm, with none appearing within the vacuole (*Figure 1C*, Non-starvation). A similar cytoplasmic location was observed for LDs in cells undergoing gradual GR (*Figure 1D* and *Video 1*, Gradual GR). By contrast, in cells starved by acute GR, LDs were observed in both the cytoplasm and vacuole (*Figure 1D* and *Video 2*, Acute GR); those in the vacuole appeared to have been engulfed as whole droplets. Quantification of the images revealed that approximately 60% of cells starved by acute GR contained vacuole-localized LDs compared to only ~10% of cells starved by gradual GR and none of non-starved cells (*Figure 1E*). These results suggested that under acute GR cells quickly activate a cellular response pathway involving LD delivery into the vacuole, where LDs are digested to fuel mitochondrial OXPHOS.

## Vacuole uptake of LDs under acute GR occurs by μ-lipophagy

LD delivery to and degradation in the vacuole could occur either by an autophagy -dependent or -independent pathway (*van Zutphen et al., 2014*; *Wang et al., 2014*). Since autophagy requires core machinery, including Atg1p and Atg8p (*Mizushima et al., 2011*; *Parzych and Klionsky, 2014*), we monitored the location and quantity of LDs using SXT in autophagy-deficient *atg1Δ* and *atg8Δ*

cells under acute or gradual GR conditions. Virtually, no LDs were found inside the vacuole of *atg1Δ* or *atg8Δ* cells at 1.5 days of acute GR, in contrast to wild-type (WT) cells undergoing acute GR (*Figure 2A*, top panel, see quantification in *Figure 2B*). Instead, the droplets were primarily localized in the cytoplasm, similar to that seen under gradual GR conditions (*Figure 2A*, bottom panel). This indicated that LD accumulation within the vacuole during acute GR occurs by autophagy. During this process, LDs appeared to be directly engulfed by the vacuole by a micro-autophagy mechanism, as previously reported (*van Zutphen et al., 2014*; *Vevea et al., 2015*). Herein, we call this process μ-lipophagy.

Several lines of evidence suggested that LD content delivered into the vacuole by μ-lipophagy was catabolized there for energy production. First, WT cells undergoing acute GR had ~50% less total LD volume relative to the cells undergoing gradual GR, with most LD volume in the vacuole; and, this reduction in total LD volume was absent in *atg1Δ* cells undergoing acute GR (*Figure 2C*). Second, Western blot

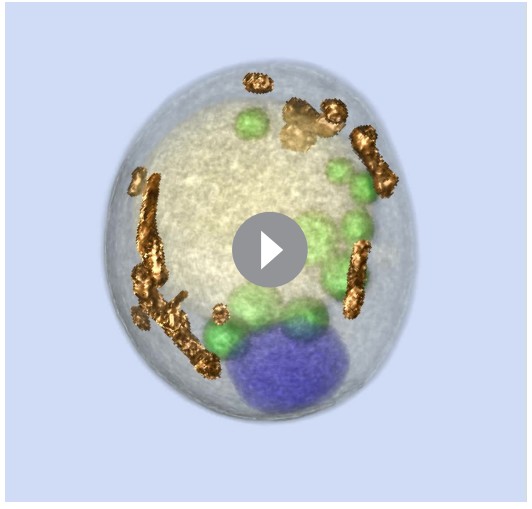

**Video 1.** A representative SXT movie of yeast cells under gradual GR. SXT images of cells grown under gradual GR for 1.5 days were processed and segmented to show individual organelle morphologies (LDs: green; nucleus: purple; vacuole: pale yellow; mitochondria: gold).

analysis of cells expressing the LD marker Erg6-DsRed revealed that within 3 days of starvation, WT cells under acute GR produced high levels of free DsRed, indicative of LD turnover, whereas similarly starved *atg1Δ* cells did not (*Figure 2D*, Acute GR). Virtually, no free DsRed was released from cells undergoing gradual GR in WT or *atg1Δ* cells (*Figure 2D*, Gradual GR). Finally, confocal imaging of Erg6-DsRed-expressing cells showed DsRed signal dispersed inside the vacuole between day 1 and day 3 of acute GR (*Figure 2E*, see arrowheads), whereas DsRed signal was associated with cytoplasmic puncta in either *atg1Δ* cells or cells starved by gradual GR. Thus, the μ-lipophagy triggered by acute GR leads to LD turnover in these cells.

Examining overall autophagy rates using GFP release from GFP-Atg8 revealed that autophagic flux was maintained long-term under acute GR but not under gradual GR (*Figure 2F*). It is cells that sense acute GR, therefore, that can initiate and maintain both μ-lipophagy and bulk autophagy pathways for long-term survival under complete nutrient deprivation.

## TOR pathway inhibition obstructs rather than boosts μ-lipophagy

To begin dissecting the mechanism(s) that trigger and maintain μ-lipophagy seen under acute GR, we asked whether amino acid sensing could play a role. To address this possibility, we replaced SD media (which lacks a majority of amino acids) with synthetic complete (SC) media, which contains excess amino acids. No significant change in the management of autophagy and LDs under acute or gradual GR occurred by this media switch (*Figure 3A and B*). Total autophagic flux and μ-lipophagy remained higher under acute GR compared to gradual GR. This suggested amino acid depletion is not a major factor in triggering μ-lipophagy during acute GR.

To further test this inference, we included rapamycin (which directly inhibits target of rapamycin (TOR), the major amino acid sensor in cells (*Abeliovich et al., 2000*; *Kamada et al., 2000*) in our experiments. We began by investigating gradual GR. Rapamycin treatment of cells undergoing gradual GR led to a transient increase in autophagic flux (*Figure 3C*, gradual GR). This was as expected given inactivated TOR's role in triggering autophagy. However, the autophagy could not be maintained and declined within 1 day. Notably, LD degradation (assessed using Erg6-DsRed) in rapamycin-treated cells under gradual GR remained low (*Figure 3D*, Gradual GR), similar to cells under gradual GR not treated with rapamycin (see Gradual GR in *Figure 3B*). This showed that TOR inactivation by rapamycin in cells undergoing gradual GR does not induce long-term autophagic flux or μ-lipophagy.

We next examined the effect of rapamycin treatment on cells undergoing acute GR. No substantial boost occurred in general autophagic flux (*Figure 3C*, Acute GR) compared to untreated cells undergoing acute GR (see *Figure 3A*, Acute GR). However, LD degradation was surprisingly decreased (*Figure 3D*, Acute GR) relative to untreated cells undergoing acute GR (see *Figure 3B*, Acute GR). This suggested that inactivation of the amino-acid-sensing TOR system during acute GR blocks μ-lipophagy rather than promotes it.

## AMPK involvement in μ-lipophagy and cell survival under acute GR

AMPK, the master energy sensor within cells, is activated by glucose depletion (*Cantó and Auwerx, 2011*). We tested its role in inducing μ-lipophagy in cells undergoing acute GR by examining starved cells lacking AMPK (i.e. *snf1Δ* or *snf4Δ* cells) (*Jiang and Carlson, 1996*). The high levels of LD degradation (assessed by release of free DsRed from Erg6-DsRed) characteristic of acute GR in WT cells were not seen in *snf1Δ* or *snf4Δ* cells undergoing acute GR (*Figure 3E*, Acute GR), and instead resembled that of WT, *snf1Δ*, or *snf4Δ* cells when undergoing gradual GR (*Figure 3E*, Gradual GR). LD consumption seen under acute GR thus appears to require AMPK activity.

Several lines of evidence suggested that the defect in LD consumption in AMPK-deficient cells occurs at the level of initiation of μ-lipophagy. When *snf1Δ* and *snf4Δ* cells were labeled with Erg6-DsRed and Vph1-GFP (a vacuole membrane marker) and starved by acute GR for 3 days, no DsRed signal inside the vacuole was seen by confocal imaging, in contrast to similarly-starved WT cells (*Figure 3F*). Examination of *snf1Δ* cells by SXT after 1.5 days of acute GR revealed LDs were excluded from the vacuole, in contrast to WT cells undergoing acute GR, where LDs were readily taken up (*Figure 3G*, black arrows point to LDs). Examining full 3D SXT tomograms showed that LDs in *snf1Δ* cells had an overall larger size and that the total LD volume in these cells was increased

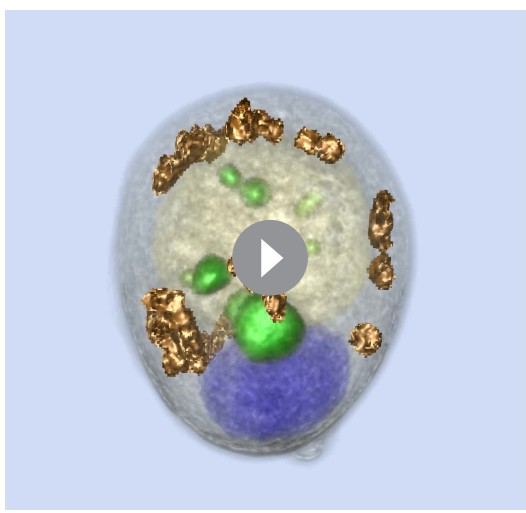

**Video 2.** A representative SXT movie of yeast cells under acute GR. SXT images of cells grown under acute GR for 1.5 days were processed and segmented to show individual organelle morphologies (LDs: green; nucleus: purple; vacuole: pale yellow; mitochondria: gold).

relative to WT cells undergoing acute GR (*Figure 3—figure supplement 1A and B*), as expected if μ-lipophagy was not induced.

Autophagic bodies (ABs), but not LDs, could be seen inside the vacuole in SXT images of *snf1Δ* cells starved by acute GR (*Figure 3G*, white arrows point to these ABs, which are less dense than LDs because they contain fluid portions of the cytoplasm with less lipid [*Tsukada and Ohsumi, 1993*]). This suggested that μ-lipophagy, not bulk autophagy, is preferentially inhibited when AMPK is absent from cells starved by acute GR. Supporting this, total autophagic flux (measured using GFP release from GFP-Atg8) decreased but was not completely abolished in *snf1Δ* or *snf4Δ* cells undergoing acute GR relative to similarly starved WT cells (*Figure 3H* and *Figure 3—figure supplement 1C*).

In addition to μ-lipophagy being blocked in AMPK-deficient cells (i.e. *snf1Δ* cells), these cells had significantly reduced ATP levels and shortened lifespans under acute GR, resembling cells undergoing gradual GR (*Figure 3—figure supplement 1D and E*). Under acute GR, therefore, AMPK activation, μ-lipophagy, and downstream ATP generation appear to be tightly linked for ensuring cell survival.

## AMPK regulation under glucose starvation

To analyze how AMPK is regulated under acute versus gradual GR, we examined the extent to which Snf1p is phosphorylated to become active under these conditions. Measuring Thr210 phosphorylation of Snf1p as indicator of AMPK activation (*McCartney and Schmidt, 2001*), we found AMPK became active within 4 hr of acute GR, whereas non-starved cells or cells undergoing gradual GR showed little AMPK activation at this time (*Figure 4A*). AMPK activation also remained low under gradual GR conditions when cells were incubated for 4 hr in media with rapamycin. This suggests that TOR inhibition also does not activate AMPK at this time point (*Figure 4B*).

We next examined Thr210 phosphorylation of Snf1p after longer times of acute or gradual GR (*Figure 4C and D*). Under acute GR, phosphorylated Snf1p levels measurably increased after 18 hr and were maintained thereafter. Under gradual GR, by contrast, significant levels of phosphorylated Snf1p were present at 18 hr, but the levels fell thereafter, becoming very low by 36 hr (*Figure 4C*). Notably, addition of the proteasome inhibitor MG132 to these cells at 18 hr increased total Snf1p levels at 24 hr and 36 hr (*Figure 4C*), although by 36 hr the phosphorylated pool of Snf1p in total Snf1p was still decreased in cells undergoing gradual GR (*Figure 4D*). This suggested that AMPK is activated only short-term under gradual GR because the protein and its phosphorylated form undergo rapid proteasomal degradation. By contrast, under acute GR, Snf1p and its phosphorylated form appear to be protected from degradation and therefore more stable.

Interestingly, loss of phosphorylated and total Snf1p was accelerated within 24 hr of acute GR when TOR was inhibited by rapamycin treatment (*Figure 4E*, Acute GR). Degradation of phosphorylated and total Snf1p also sped up during gradual GR when rapamycin was present, with Snf1p no longer detectable at 24 hr treatment (*Figure 4E*, Gradual GR). This indicated rapamycin treatment antagonizes AMPK activity. As μ-lipophagy depends on AMPK activation (see *Figure 3E*, Acute GR), this could explain why rapamycin treatment decreases μ-lipophagy under acute GR (see *Figure 3D*).

We next examined the effect of rapamycin treatment on cell survival under acute or gradual GR. Rapamycin treatment diminished cell viability in a dose-dependent manner under both acute and gradual GR conditions (*Figure 4F* and *Figure 4—figure supplement 1A*). A similar effect was observed when amino acids were depleted from the media (i.e., SD gradual and SD acute GR)

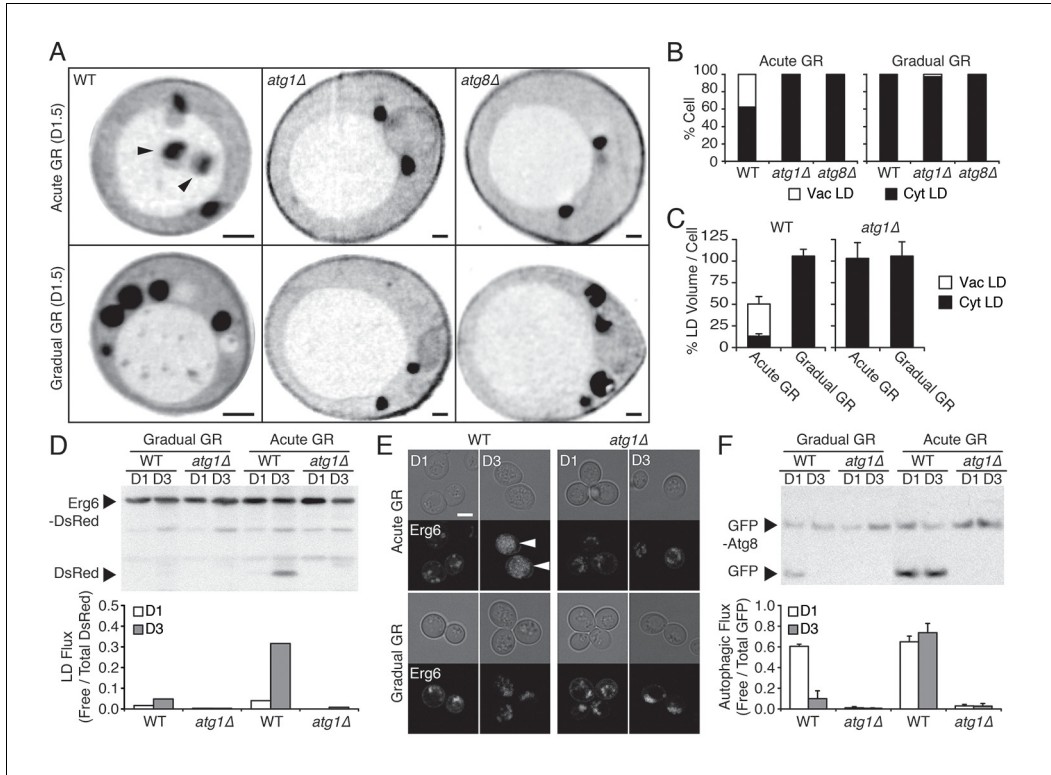

**Figure 2.** Under acute GR, cytosolic LDs are engulfed by the vacuole and digested via μ-lipophagy. (A) Representative SXT orthoslice images of WT, *atg1Δ*, and *atg8Δ* cells undergoing acute or gradual GR for day 1.5 (D1.5) are shown. Black arrowheads point to LDs inside the vacuole. Scale bar represents 0.5 μm. (B) Percentage of cells having only cytoplasmic LDs (Cyt LD) or having both Cyt LD and vacuole associated LDs (Vac LD) are shown. Data were analyzed from full 3D tomograms of the SXT images in (A). Approximately 20 cells per each condition were analyzed. (C) LD volume analyzed from full 3D SXT images of WT and *atg1Δ* cells in (A) is shown as a percentage of that seen in cells undergoing gradual GR (which was set at 100%). Data are expressed as mean ± SD (*n* = 7 ~ 10). (D) WT and *atg1Δ* cells expressing Erg6-DsRed (LD marker) were grown under similar conditions in (A) and Erg6p degradation was measured by Western blot analysis. Quantification of Erg6p degradation is shown in the lower panel as ratio of free DsRed to total DsRed (free DsRed + Erg6-DsRed) signals. (E) Representative confocal images of cells in (D) are shown. White arrowheads point to vacuolar localized LD signals. Scale bar represents 5 μm. (F) WT and *atg1Δ* cells expressing GFP-Atg8 (autophagosome marker) were grown under similar conditions in (A) and Atg8p degradation was measured by Western blot analysis. Quantification of Atg8p degradation is shown in the lower panel as ratio of free GFP to total GFP (free GFP + GFP-Atg8) signals. Data are expressed as mean ± SD (*n* = 3). n indicates the number of experimental repetitions.

(*Figure 4—figure supplement 1B*). Therefore, TOR inactivation is insufficient to keep cells alive under glucose restriction, potentially because it inhibits long-term AMPK activation, which is necessary to maintain μ-lipophagy.

## Testing for autophagic machinery specific for μ-lipophagy

We next investigated whether specific autophagy factors initiate μ-lipophagy in cells undergoing acute GR. Parallel GFP-Atg8 and Erg6-DsRed flux assays (measuring bulk and LD-specific degradation, respectively) were performed in yeast cells lacking each of the 31 known autophagy-related (*ATG*) genes under acute GR (*Figure 5A* and *Figure 5—figure supplement 1*). Thirteen *ATG* genes known to be essential for general autophagy were also necessary for μ-lipophagy; including genes for autophagosome induction (i.e. *ATG1, ATG6, ATG13* and *ATG14*), maturation (i.e. *ATG3, ATG5, ATG7, ATG10, ATG12* and *ATG16*), and membrane retrieval (i.e. *ATG2, ATG9 and ATG18*) (*Nakatogawa et al., 2009*) (*Figure 5—figure supplement 1*). This indicated that μ-lipophagy in cells undergoing acute GR depends on *bona fide* autophagosomal machinery.

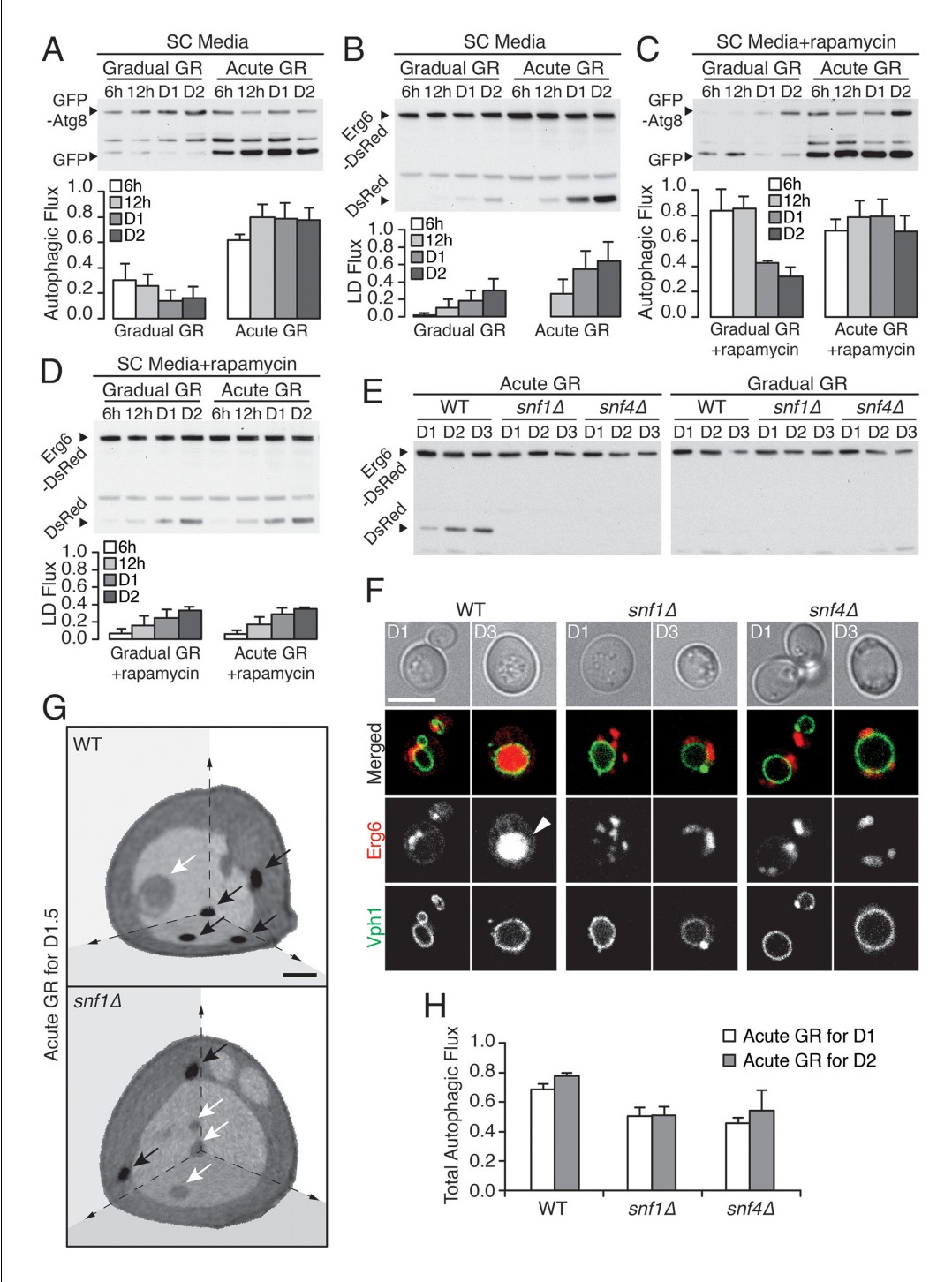

**Figure 3.** Vacuolar LD consumption is dependent on AMPK and is not induced by rapamycin treatment. (A) WT cells expressing GFP-Atg8 were placed in synthetic complete (SC) media containing 2% glucose (Gradual GR) or 0.4% glucose (Acute GR). Autophagic flux assay was performed as described in *Figure 2F*. (B) LD flux assay as described in *Figure 2D* was performed under similar conditions as in (A). (C) Autophagic flux assay was performed with 50 nM rapamycin under similar conditions as described in (A). (D) LD flux assay was performed with 50 nM rapamycin under similar conditions as described in (B). (E) LD flux assay was performed in WT, *snf1Δ*, and *snf4Δ* cells as described in *Figure 2D*. (F) Representative confocal images of cells in (E) with both Erg6-DsRed and Vph1-GFP (vacuole membrane marker) are shown. White arrowhead points to LD signal localized within the vacuole. Scale bar represents 5 μm. (G) Representative SXT images of WT and *snf1Δ* cells undergoing acute GR for day 1.5 (D1.5) are shown. Black arrows and white arrows point to LDs and autophagic bodies, respectively. Scale bar represents 1 μm. (H) Quantification of autophagic flux in WT, *snf1Δ*, and *snf4Δ* cells is shown as described in *Figure 2F*. Data are expressed as mean ± SD (*n* = 3). n indicates the number of experimental repetitions.

*Figure 3 continued on next page*

*Figure 3 continued*

The following figure supplement is available for figure 3:

**Figure supplement 1.** Snf1p is required for vacuolar LD consumption, cellular energy metabolism, and long-term survival during acute GR.

We found that four out of the 31 *ATG* genes functioned in µ-lipophagy without complete loss of autophagy activity in acutely glucose-restricted cells: these were *ATG8, ATG14, ATG23,* and *ATG34* (*Figure 5—figure supplement 1*). To pare down this list, we excluded *ATG8* because exogenous GFP-Atg8 necessarily rescues otherwise defective autophagic activity in *atg8Δ* cells. We removed *ATG23* because under acute GR, *atg23Δ* cells still produced some free DsRed as seen with longer exposure of the blots (data not shown). We also excluded *ATG34* since GFP release from GFP-Pho8Δ60 (a cytosolic substrate that must undergo autophagy for degradation [*Klionsky, 2007*]) was completely blocked in *atg34Δ* cells, unlike that found in cells lacking *SNF1* or *ATG14* during acute GR (*Figure 5B*). This left as our candidate gene *ATG14* (*Figure 5A and B*).

## Atg14p's role in µ-lipophagy

Atg14p is known to be a component of phosphatidylinositol 3-kinase (PI3K) complex I, which is involved in autophagosomal membrane biogenesis. It forms a complex with Atg6p to support the lipid kinase activity of Vps34p under nitrogen starvation (*Araki et al., 2013*; *Obara and Ohsumi, 2011*). Consistent with this role, we observed much less autophagic flux (*Figure 5—figure supplement 2A*) and fewer Atg8 punctate structures (*Figure 5—figure supplement 2B and C*) in *atg14Δ* cells relative to WT cells under nitrogen starvation. Additionally, when we treated cells expressing GFP-Atg8 with rapamycin and measured autophagic flux in WT, *atg1Δ*, or *atg14Δ* cells, autophagic flux was inhibited in both *atg1Δ* and *atg14Δ* cells relative to WT cells (with less inhibition in *atg14Δ* compared to *atg1Δ* cells) (*Figure 5—figure supplement 2D*).

In contrast to nitrogen starvation and rapamycin treatment, Atg14p was not essential for general autophagic flux under acute GR. GFP release from GFP-Atg8 (*Figure 5A*, GFP-Atg8 panel) or GFP-Pho8Δ60 (*Figure 5B*) still occurred in *atg14Δ* cells, decreasing only slightly compared to WT cells. Despite this, *atg14Δ* cells undergoing acute GR showed a significant block in LD degradation (*Figure 5A*, Erg6-DsRed panel), implying Atg14p's role in cells undergoing acute GR is to drive µ-lipophagy.

We tested this hypothesis with confocal imaging of Erg6-DsRed and direct visualization of LDs using SXT in *atg14Δ* cells under acute GR. No Erg6-DsRed or LD structures redistributed into the vacuole in these cells (*Figure 5C and D*), supporting a role of Atg14p in initiating µ-lipophagy. By contrast, Atg14p was not required for pexophagy, the specialized form of autophagy involved in degrading peroxisomes (*Suzuki, 2013*): substantial degradation of the peroxisomal marker Pot1-GFP still occurred in *atg14Δ* cells under GR, whereas no Pot1-GFP degradation occurred in *atg11Δ* cells, where pexophagy is blocked (*Figure 5—figure supplement 2E and F*). Given that *ATG14* has also been shown to be dispensable for autophagic degradation of ER components (*Schuck et al., 2014*), we concluded Atg14p's role under acute GR is primarily to induce µ-lipophagy.

## Cellular impact of *ATG14* deletion

We next examined the effect of deleting *ATG14* on LD consumption rates, cellular ATP levels, and cell survival in cells undergoing acute GR. We found that *atg14Δ* cells under acute GR: had less LD volume loss relative to that of WT or *atg11Δ* cells under acute GR (*Figure 5E*); depleted their ATP levels significantly faster than WT cells (or pexophagy-deficient *atg11Δ* cells) under acute GR (*Figure 5F*); and had significantly shortened lifespans relative to WT or *atg11Δ* cells under acute GR (*Figure 5G*). The shortened lifespan of *atg14Δ* cells under acute GR was similar to that seen for *atg1Δ, atg5Δ,* and *atg23Δ* cells, which lack core general autophagy machinery (see *Figure 5—figure supplements 1* and *3A*). Therefore, Atg14p and other core autophagy machinery are critical for the energy metabolism and long-term survival of cells starved acutely of glucose. Other types of autophagy, including pexophagy (controlled by Atg11p) and mitophagy (controlled by Atg11p, Atg32p, and Atg33p), did not have this role (*Figure 5G* and *Figure 5—figure supplement 3B and C*).

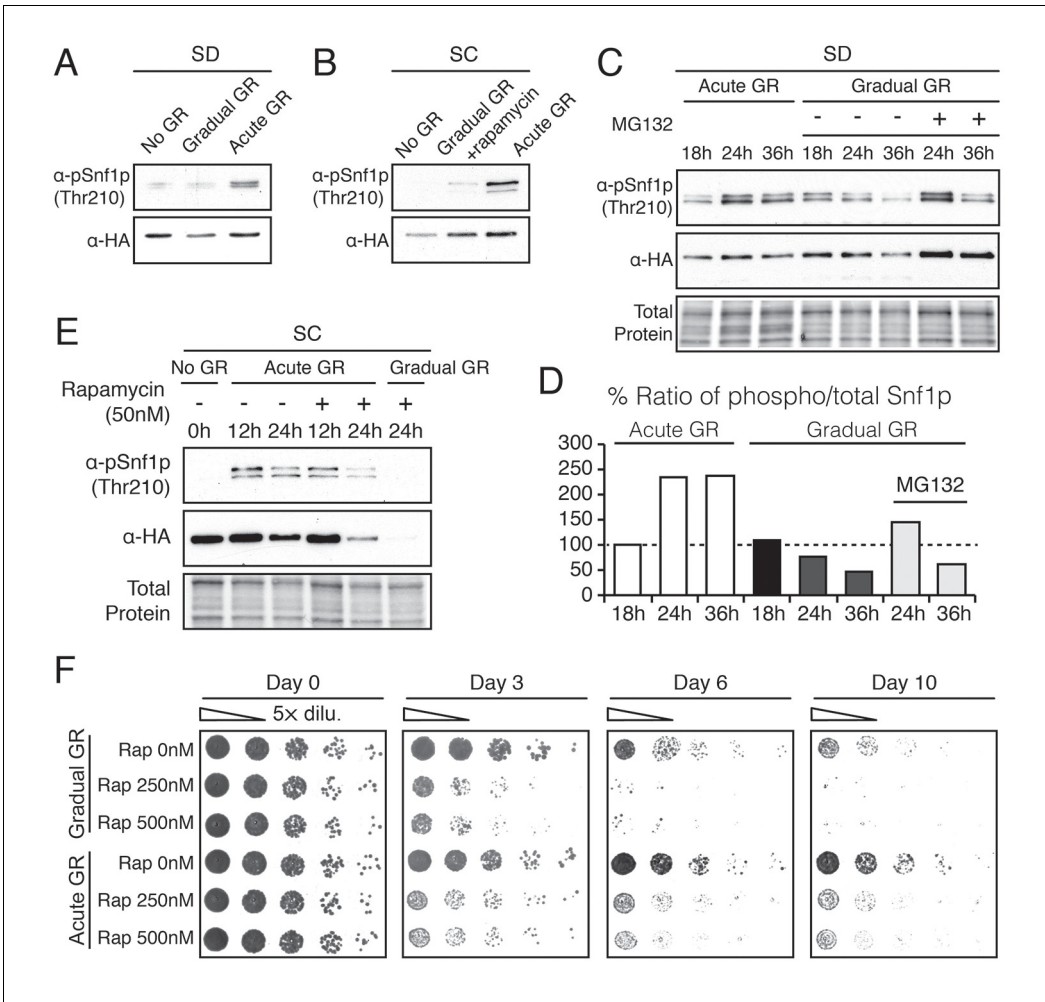

**Figure 4.** Acute GR, but not gradual GR or rapamycin treatment, enhances AMPK activity to promote cell survival. (**A**) Phosphorylated or total Snf1p levels were determined in cells expressing Snf1-HA under no GR and gradual or acute GR for 4 hr using α-phospho AMPK (top band: HA-tagged Snf1p; lower band: endogenous Snf1p) and α-HA antibodies, respectively. (**B**) Cells harboring Snf1-HA were placed in synthetic complete (SC) media containing 2% glucose with 50 nM rapamycin (Gradual GR +rapamycin) or 0.4% glucose (Acute GR) for 4 hr. Phosphorylated or total Snf1p levels were measured as described in (**A**). (**C**) MG132 (10 μM at final concentration) was added to cells in (**A**) and the levels of phosphorylated or total Snf1p were determined as described in (**A**) at the indicated times. (**D**) The ratio of phosphorylated to total Snf1p levels was determined using data in (**C**) and was plotted as a percentage of that seen in cells undergoing acute GR at 18 hr (which was set at 100%). (**E**) Cells harboring Snf1-HA were grown under SC media with no GR, acute GR, or gradual GR with/without rapamycin treatment. The levels of phosphorylated or total Snf1p were measured as described in (**A**) at the indicated times. (**F**) Cell survival assay was performed in WT cells undergoing gradual or acute GR in the presence of rapamycin (0, 250, and 500 nM at final concentration) in SC media. Equal numbers of cells from each condition were plated onto YPEG agar media to visualize viable cells at the indicated days. Representative colony images are shown.

The following figure supplements are available for figure 4:

**Figure supplement 1.** TOR inhibition by rapamycin treatment or amino acid depletion obstructs cell survival during glucose starvation.

**Figure supplement 2.** Constitutively-active AMPK does not extend lifespan of cells undergoing gradual GR.

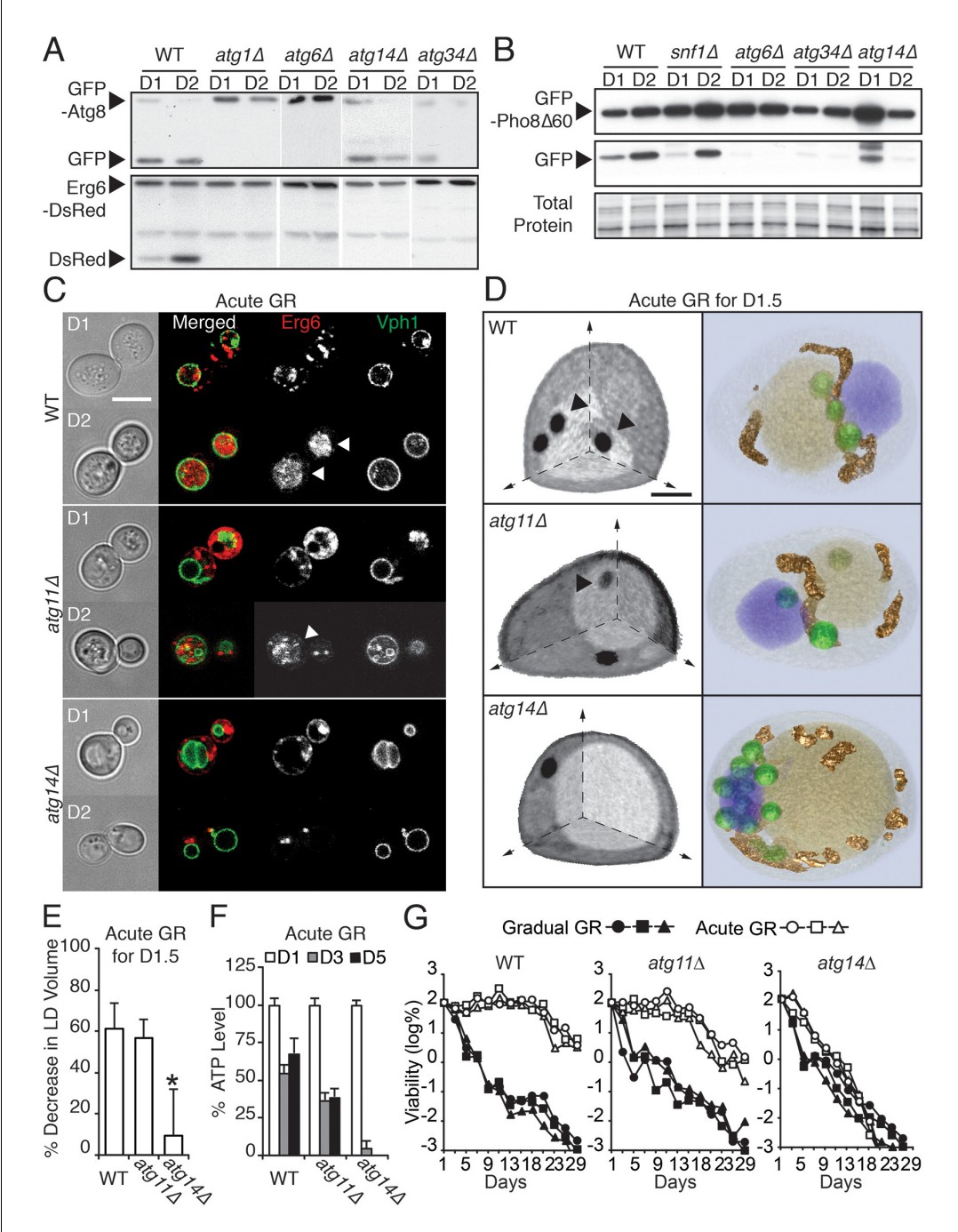

**Figure 5.** *ATG14* is required for µ-lipophagy, ATP maintenance, and lifespan extension during acute GR. (A) Autophagic and LD flux assays were performed as described in *Figure 2F and D*, respectively. Western blot images of WT, *atg1Δ*, *atg6Δ*, *atg14Δ*, and *atg34Δ* cells grown under acute GR are shown. (B) Pho8Δ60 (a cytoplasmic autophagic substrate) degradation assay was performed in WT, *snf1Δ*, *atg6Δ*, *atg14Δ*, and *atg34Δ* cells harboring GFP-Pho8Δ60 during acute GR. (C) Representative confocal images of WT, *atg11Δ*, and *atg14Δ* cells expressing Erg6-DsRed and Vph1-GFP under acute GR are shown. White arrowheads point to vacuolar localized LD signals. Scale bar represents 5 µm. (D) Representative SXT images of WT, *atg11Δ*, and *atg14Δ* cells are shown. Black arrowheads point to LDs localized within the vacuole. Scale bar represents 1 µm. Right panels show full 3D SXT images (LD: green; nucleus: purple; vacuole: pale yellow; mitochondria: gold). (E) Percentage decrease of LD volume in acute GR compared to gradual GR was determined using full 3D SXT images of cells in (D). Data are expressed as mean ± SD (*n* = 7 ~ 10). *p<0.05 versus WT. (F) ATP levels of cells in (E) were determined as described in the Materials and methods. Data are expressed as mean ± SD (*n* = 3). n indicates the number of experimental repetitions. (G) Lifespan of cells in (E) undergoing acute or gradual GR is shown as described in *Figure 1A*. Three biologically independent experiments are shown together.

*Figure 5 continued on next page*

*Figure 5 continued*

The following figure supplements are available for figure 5:

**Figure supplement 1.** Biochemical screening reveals differential roles of autophagy-related (*ATG*) genes in μ-lipophagy.

**Figure supplement 2.** *ATG14* is required for autophagy induction, while its deficiency does not block degradation of autophagosome or peroxisome during glucose starvation.

**Figure supplement 3.** Genes required for μ-lipophagy, but not for pexophagy or mitophagy, are necessary for cell longevity during acute GR.

Under gradual GR (in which μ-lipophagy is not induced), *atg14Δ* cells displayed an almost identical lifespan as WT or *atg11Δ* cells, with the vast majority of these cells dead by 13 days (>99.9%, *Figure 5G*). This suggested Atg14p's pro-survival role in starved cells is connected to its regulation of μ-lipophagy.

## Mechanism(s) underlying Atg14p's regulation of μ-lipophagy

To gain insight into how μ-lipophagy induction under acute GR is regulated by Atg14p, we monitored changes in Atg14p distribution under different GR conditions. Under no GR or gradual GR, Atg14p was localized solely within the cytoplasm in a dispersed and/or punctate pattern (*Figure 6A and B*, and *Figure 6—figure supplement 1*). The punctate structures co-localized with Sec13p (a subunit of the COPII vesicle coat mediating ER-to-Golgi transport) (*Figure 6C*), supporting prior work implicating Atg14p in the formation of pre-autophagosomal structures at ER exit sites (ERES) under nitrogen starvation conditions (*Graef et al., 2013*). In contrast to this distribution, Atg14p in cells undergoing acute GR became highly enriched on the vacuole surface (*Figure 6A–C*). Notably, Atg14p failed to redistribute onto the vacuole surface and remained at ERES in *snf1Δ* cells undergoing acute GR (*Figure 6D*), in which AMPK is absent and μ-lipophagy is inhibited. Re-introduction of Snf1p-HA into *snf1Δ* cells under acute GR led to Atg14p re-localizing onto the vacuole surface (*Figure 6A and B*), and reversed the growth defects seen in these cells under glucose starvation (*Figure 6—figure supplement 2*). These results suggested that under acute GR, Atg14p redistributes in an AMPK-dependent manner from ERES to vacuolar membranes where it helps to induce μ-lipophagy. Atg14p appeared to function downstream of AMPK in this pathway because *atg14Δ* cells did not prevent AMPK (i.e. Snf1p) from being phosphorylated under acute GR (*Figure 6—figure supplement 3A*), whereas *snf1Δ* cells prevented Atg14p from re-localizing onto the vacuole (see above).

Given that rapamycin treatment under acute GR inhibits μ-lipophagy, we next looked at Atg14p localization under these conditions. Notably, Atg14p did not relocate onto the vacuolar surface in rapamycin-treated cells undergoing acute GR (*Figure 6E–G*); instead, it localized within cytoplasmic puncta that often co-localized with Sec13p (see arrowheads in *Figure 6F*). Thus, Atg14p's localization to the vacuole under acute GR is antagonized by TOR inhibition (i.e. rapamycin treatment).

Immunoprecipitation experiments using cells harboring a Snf1p-HA plasmid and GFP-tagged Atg14p revealed Atg14p interacts with Snf1p-HA upon acute GR, in contrast to another autophagy-related PI3K component, Atg6p (*Obara and Ohsumi, 2011*), which did not significantly interact with Snf1p compared to Atg14p (*Figure 6—figure supplement 3B*). AMPK and Atg14p therefore appear to coexist in a complex under acute GR conditions.

## Testing for vacuolar localization of other autophagy-related PI3K complex I components

We examined whether other autophagy-related PI3K complex I components besides Atg14p (i.e., Atg6p, Atg38p, Vps15p, Vps34p, and Vps38p) relocated to the vacuole surface under acute GR to help initiate μ-lipophagy. Cells harboring Ivy1p-mCherry (a vacuole membrane marker) and GFP-tagged Atg6p, Atg38p, Vps15p, Vps34p, or Vps38p were monitored. Vacuole association of each PI3K complex I component was scored by AIRY-scan confocal microscopy image analysis. Prior to glucose starvation, Atg6p and Vps15p both showed significant association with the vacuole surface.

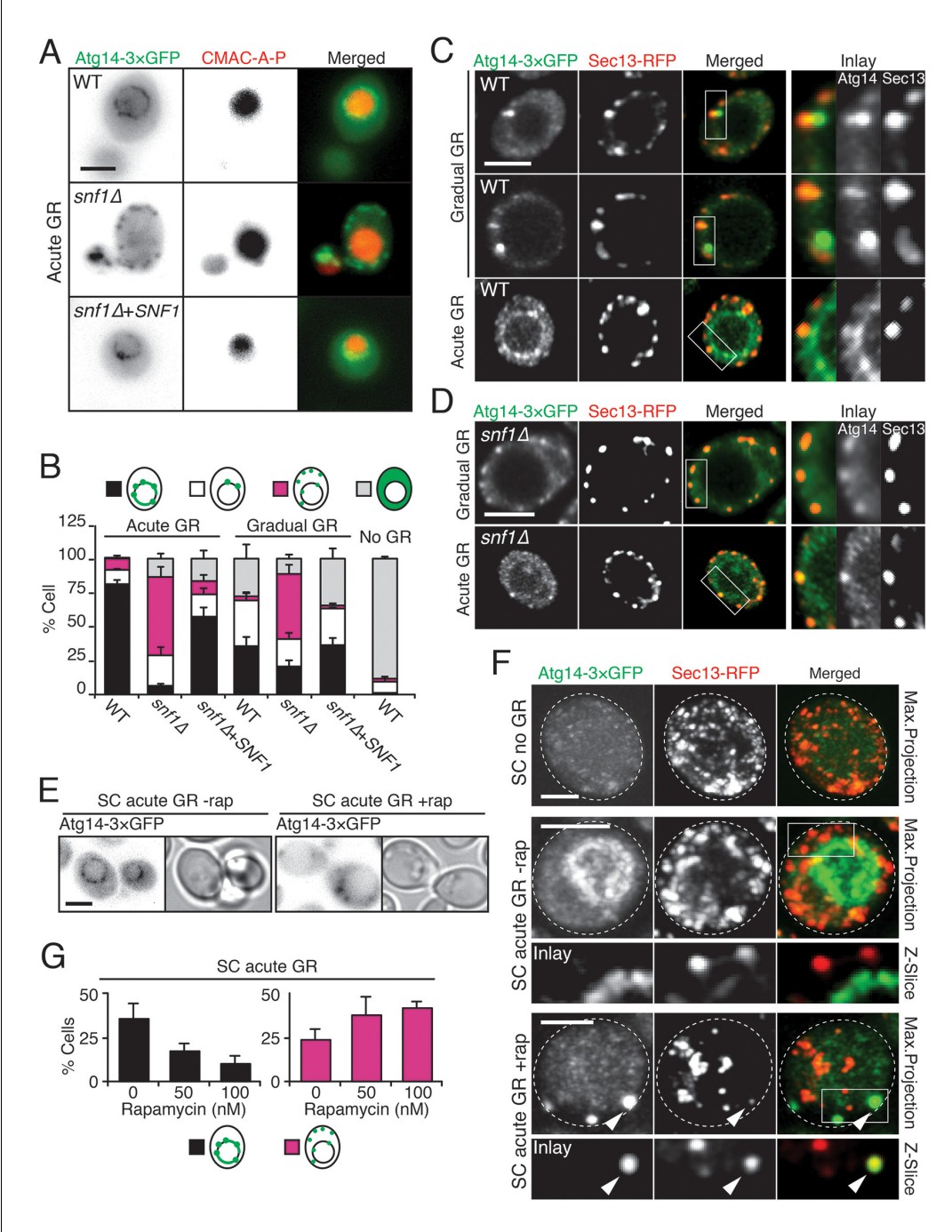

**Figure 6.** Atg14p is enriched on the vacuole surface upon AMPK activation during acute GR. (**A**) WT and *snf1Δ* cells harboring Atg14-3×GFP were grown under no GR, gradual GR, or acute GR for ~24 hr with or without Snf1p-HA. Representative images of Atg14p and the vacuole marker CMAC-A-P are shown. (**B**) Percentage of cells in (**A**) displaying indicated Atg14p localization patterns was determined. Approximately 100 cells per each condition were analyzed. (**C**) Cells harboring Atg14-3×GFP and Sec13-RFP were grown under gradual or acute GR for ~24 hr. Representative images of Atg14p and Sec13p are shown. (**D**) *snf1Δ* cells harboring Atg14-3×GFP and Sec13-RFP were grown under gradual or acute GR for ~24 hr. Representative images of Atg14p and Sec13p are shown. (**E**) Representative images of Atg14p in cells undergoing acute GR with excessive amino acids (i.e. SC acute GR) with/without 50 nM rapamycin (rap) for ~24 hr are shown. (**F**) Representative images of Atg14p and Sec13p in cells growing under excessive amino acids with no GR (i.e. SC no GR) or acute GR (i.e. SC acute GR) with/without 50 nM rap for ~24 hr are shown. Dashed lines and arrowheads indicate cell membrane and Atg14p signals that are co-localized with Sec13p, respectively. (**G**) Percentage of cells in (**E**) displaying indicated Atg14p localization patterns is shown. Approximately 100 cells per each condition were analyzed. Unless indicated otherwise, data from five independent measurements are shown as mean ± SD. Scale bar represents 2.5 μm.

*Figure 6 continued on next page*

*Figure 6 continued*

The following figure supplements are available for figure 6:

**Figure supplement 1.** Atg14p is localized within the cytoplasm in dispersed or punctate pattern under no GR or gradual GR.

**Figure supplement 2.** Reintroducing *SNF1*-3×HA rescues growth defects of *snf1Δ* cells under glucose starvation.

**Figure supplement 3.** Atg14p forms a complex with Snf1p and is not required for Snf1p phosphorylation during acute GR.

Whereas Atg6p remained localized there after shifting to either acute or gradual GR, Vps15p redistributed off the vacuole upon acute GR (*Figure 7A and B*). The other PI3K complex I components (i.e. Atg38p, Vps34p, and Vps38p) remained primarily cytosolic with no increased vacuolar

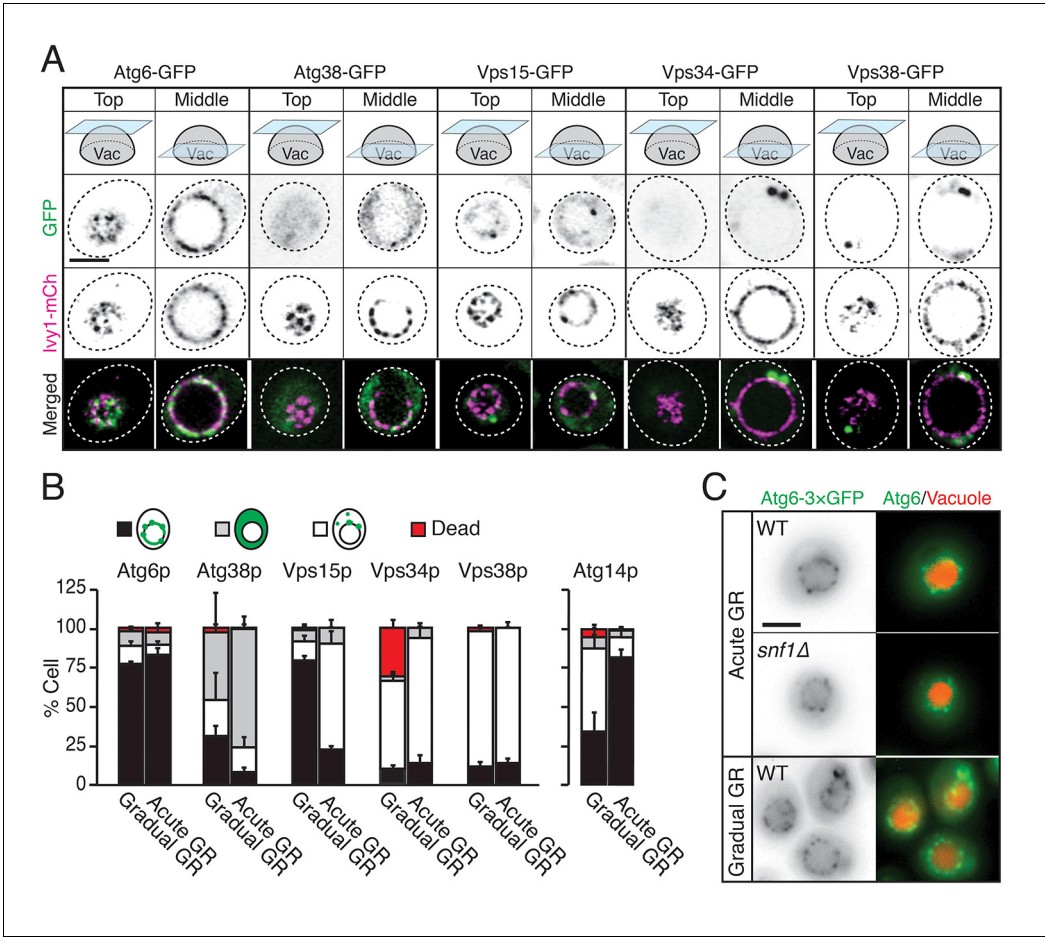

**Figure 7.** Atg6p and Atg14p, but not other PI3K complex I components, enrich on the vacuole surface upon acute GR. (**A**) Cells harboring GFP-tagged Atg6p, Atg38p, Vps15p, Vps34p, or Vps38p and containing Ivy1-mCherry (Ivy1-mCh, vacuole membrane marker) were grown under acute or gradual GR. Representative images of each indicated GFP-tagged protein and Ivy1p in cells under acute GR for ~24 hr are shown. (**B**) Cells in (**A**) or containing Atg14-3×GFP were grown under gradual or acute GR for ~24 hr. Percentage of cells displaying indicated GFP localization patterns was determined. Approximately 100 cells per each condition were analyzed. Data are expressed as mean ± SD (*n* = 3). n indicates the number of experimental repetitions. (**C**) WT and *snf1Δ* cells harboring Atg6-3×GFP were grown under acute or gradual GR for ~24 hr. Representative images of Atg6p and the vacuole marker CMAC-A-P are shown. Scale bar represents 2.5 μm.

association upon shift to either acute or gradual GR (*Figure 7A and B*). This meant that only Atg6p and Atg14p reside on the vacuole surface under acute GR, potentially serving as important initiators of μ-lipophagy. In addition, only Atg14p's vacuolar localization was AMPK-dependent, since Atg6p remained prominently localized at the vacuolar surface even when AMPK was inactive (i.e. in *snf1Δ* cells or during gradual GR) (*Figure 7C*).

## Atg14p drives liquid-ordered membrane domain formation on the vacuole

Prior work has shown that when budding yeast are starved of glucose, they form distinct sterol-enriched, liquid-ordered ($L_o$) domains and liquid-disordered membrane ($L_d$) domains on their vacuolar surface (*Toulmay and Prinz, 2013*), with the $L_o$ domains serving as preferential internalization sites for LDs (*Wang et al., 2014*). Using GFP-Pho8 as a marker for $L_d$ domains, we observed large, marker-free $L_o$ domains on the vacuole membrane in cells undergoing acute GR (*Figure 8A*). LDs, labeled using Erg6-DsRed, resided exclusively in the $L_o$ domains, consistent with the $L_o$ domains serving as preferential sites for LD uptake into the vacuole. By contrast, in cells undergoing gradual GR, $L_o$ domains were much less apparent, and LDs were mainly clustered near one side of the vacuole (*Figure 8B*). Acute GR therefore appears to drive both the formation of large $L_o$ domains on the vacuole membrane and LD association with these domains.

We next used AIRY-scan confocal microscopy to visualize the vacuolar localizations of Atg14p, Atg6p and a different vacuole $L_d$ domain marker, Vph1-mCherry (*Toulmay and Prinz, 2013*) under acute GR. 3D volume reconstruction and maximum projection images revealed Atg14p molecules localizing as punctate elements within vacuolar $L_o$ domains (*Video 3* and *Figure 8C*). Single z-slices from these images showed Atg14p was preferentially localized on the edges of $L_o$ domains (*Figure 8C*, Single z-slice), in close association with LDs (*Figure 8—figure supplement 1*). Atg6p also localized in $L_o$ domains at the vacuole surface but was more broadly distributed relative to Atg14p (*Figure 8D*).

To address whether Atg14p and/or Atg6p are critical for creating the large $L_o$ domains on vacuole membranes under acute GR, we deleted the gene for each protein and then examined whether or not $L_o$ domains were formed. Little or no $L_o$ domains were observed in either *atg14Δ* or *atg6Δ* cells under acute or gradual GR, with the effect most potent in *atg6Δ* cells (*Figure 8E and F*). Given that Atg6p resides on the vacuole membrane irrespective of the starvation condition (see *Figure 7C*), the results suggested Atg14p triggers large-scale $L_o$ formation under acute GR, with a critical supportive role played by Atg6p. Consistent with this, fewer large-scale $L_o$ domains formed in *snf1Δ* cells under acute GR (*Figure 8E and F*), a condition where Atg6p resides on the vacuole surface but Atg14p is not recruited there (see *Figure 6D*).

## Discussion

A key nutrient in the cell's decision whether or not to proliferate is glucose. In its presence, cells grow/divide; in its absence, cells instead switch on catabolic pathways (e.g. autophagy and fatty acid breakdown) and OXPHOS, which allow long-term survival until glucose becomes available again (*Granot and Snyder, 1991*; *Herman, 2002*; *Werner-Washburne et al., 1996*). The underlying mechanism(s) for this response, including the relevant signaling and organellar systems, is unclear yet it is fundamental for understanding cell behavior under various physiological conditions. Here, we used budding yeast as a model system to study this survival response, focusing on the key players and pathways. We found that for cells to survive long-term under full nutrient starvation, they need AMPK/Snf1p, the cell's major ATP sensor, to be activated and stabilized. This occurred when cells sensed an acute depletion of glucose at the beginning of starvation (i.e. acute GR). This induced the pathways for μ-lipophagy, general autophagy, OXPHOS and stress resistance necessary for long-term survival. In contrast, all these pathways were attenuated and cellular lifespan shortened in cells lacking AMPK activity (i.e. *snf1Δ* cells). The finding that AMPK acts as a critical upstream regulator of yeast cell survival under starvation supports prior work in *C. elegans* similarly implicating AMPK's role in lifespan extension under dietary restriction (*Greer et al., 2007*; *Schulz et al., 2007*).

Other nutrient depletion conditions besides acute GR, including rapamycin treatment or gradual GR, did not maintain AMPK activity once it was switched on, and resulted in shortened cellular lifespans. We speculate this effect arose because TOR is inhibited early on by these treatments: this

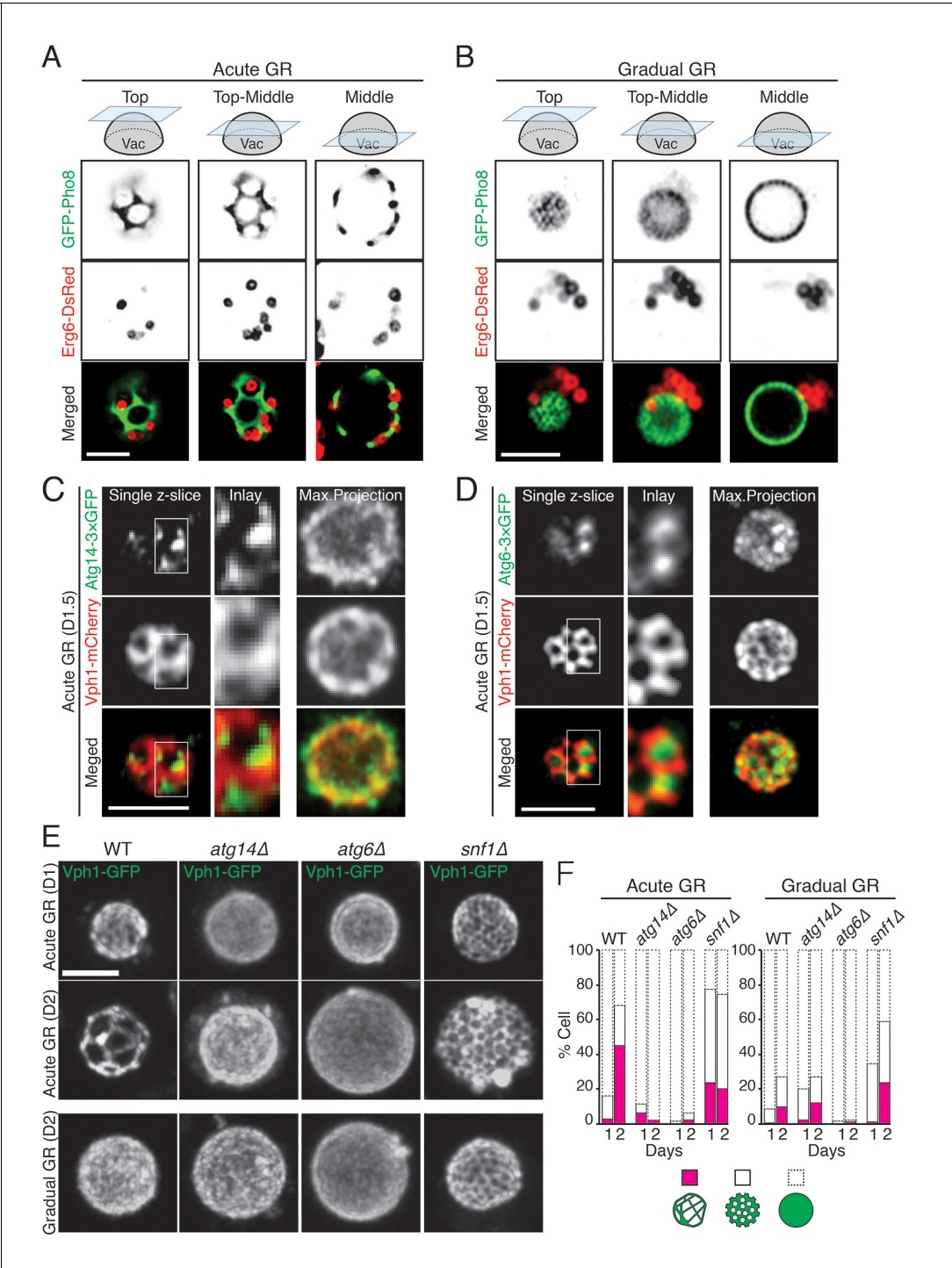

**Figure 8.** Atg14p is required for the formation of vacuolar liquid-ordered membrane domains and LD recruitment to the vacuole surface during acute GR. (A, B) Cells harboring GFP-Pho8 (a vacuole membrane marker) and Erg6-DsRed were grown under acute or gradual GR for day 1.5 (D1.5). Representative AIRY-scan images of Pho8p and Erg6p associated with different vacuole z-position are shown. (C) Cells harboring Atg14-3×GFP and Vph1-mCherry (a liquid disordered membrane maker that is excluded from liquid-ordered domains [*Toulmay and Prinz, 2013*]) were grown under acute GR for D1.5. Representative z-slice and maximum projection images of Atg14p and Vph1p are shown. (D) Cells harboring Atg6-3×GFP and Vph1-mCherry were grown under acute GR for D1.5. Representative images of Atg6p and Vph1p are shown. (E) WT, *atg14Δ*, *atg6Δ*, and *snf1Δ* cells harboring Vph1-GFP were grown under acute or gradual GR. Representative images of Vph1p at different days are shown. (F) Percentage of cells displaying indicated GFP localization patterns on the vacuole was scored. Approximately 150 cells per each condition were analyzed. Scale bar represents 2.5 μm.

*Figure 8 continued on next page*

*Figure 8 continued*

The following figure supplement is available for figure 8:

**Figure supplement 1.** Atg14p and LDs possibly interact on the vacuole liquid-ordered membrane domains upon acute GR in a Snf1p-dependent manner.

would induce bulk autophagy and cause a surplus in amino acids, raising ATP levels to antagonize AMPK activity. While rapamycin treatment directly inhibits TOR, gradual GR would do so by causing cells to sense amino acid depletion before glucose reduction from the medium, leading to TOR inhibition and its prevention of AMPK activation. This can explain why no sustained AMPK activation or enhanced longevity occurred under gradual GR, even in cells lacking Reg1p (the phosphatase that dephosphorylates Snf1p [*Bonawitz et al., 2007*]) (*Figure 4—figure supplement 2*).

Further work is needed to understand at the molecular level how TOR inhibition antagonizes AMPK activation under gradual GR. However, prior work has demonstrated that mTOR inhibition activates overall protein degradation by the ubiquitin proteasome system (*Zhao et al., 2015*), and we found that both rapamycin and gradual GR treatments caused proteasomal degradation of AMPK (in contrast to acute GR treatment, which resulted in long-term AMPK activation).

A key survival promoting role of AMPK under acute GR that we observed was the induction of μ-lipophagy, which by causing LDs to be digested provides precursors for mitochondrial energy production in the absence of glucose. In support of this role, cells undergoing acute GR with genes deleted for AMPK (i.e. *snf1Δ* cells) showed diminished LD consumption rates - comparable to those of cells undergoing gradual GR (where AMPK activation is not maintained). The conclusion that LD consumption in cells undergoing acute GR occurred by μ-lipophagy arose from two observations. First, LD consumption was autophagy-dependent; deletion of core autophagy genes blocked it (and caused diminished cell lifespan). Second, LDs were directly taken up into the vacuole, as interpreted from our results using SXT and AIRY-scan imaging and prior work using transmission EM (*Vevea et al., 2015*).

To determine what autophagic component(s) are essential for initiating μ-lipophagy under acute GR, we screened a yeast library of known autophagy genes. Under acute GR, *ATG14* was at the same time necessary for μ-lipophagy and unnecessary for general autophagy. No other gene had these characteristics, pointing to Atg14p (a subunit of the PI3K complex I) as a major component necessary for μ-lipophagy under acute GR. That said, *ATG14* was required for general autophagy under nitrogen starvation or rapamycin treatment, as previously reported (*Kametaka et al., 1998*; *Kihara et al., 2001*). Thus, there appears to be a fundamental plasticity in the regulation and function of core autophagy components under different starvation conditions- with Atg14p directing bulk autophagy under TOR inhibition, but driving μ-lipophagy under AMPK activation (i.e. acute GR).

To understand how Atg14p could be modulated to control different autophagy pathways, we examined its subcellular localization upon induction of different autophagic pathways. Under gradual GR or rapamycin treatment (where TOR is inhibited and bulk autophagy induced), Atg14p was enriched on punctate elements in the periphery of cells. These structures represented ERES, whose elements have previously been shown to recruit Atg14p in an early step in autophagosomal biogenesis (*Ge et al., 2013*; *Graef et al., 2013*). By contrast, during acute GR (where AMPK is active and μ-lipophagy is induced), Atg14p became enriched on the vacuole membrane, localizing to $L_o$ domains defined by lack of vacuolar Vph1p labeling (*Toulmay and Prinz, 2013*). Inhibiting TOR through rapamycin treatment during acute GR caused Atg14p to redistribute back to ERES, and also led to blocking of

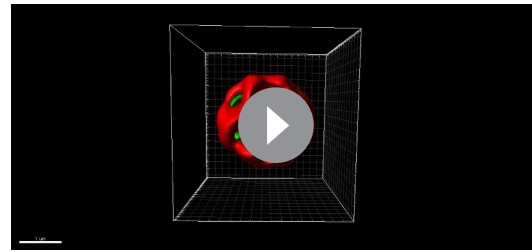

**Video 3.** Vacuolar organization of Atg14p and Vph1p during acute GR. Movie of a cell from *Figure 8C* showing 3D organization of Atg14p (green) and Vph1p (red) is shown.

μ-lipophagy. Thus, Atg14p shifts its distribution under different starvation regimens to facilitate different autophagic pathways.

What causes Atg14p's redistribution from ERES/cytosol onto the vacuolar surface upon acute GR remains unclear but could be related to the physical interaction between Snf1p and Atg14p that we observed arising under these conditions. This interaction could help stabilize Snf1p against proteasomal degradation (occurring when TOR is inactivated or under glucose-rich conditions) and may signal Atg14p's redistribution from ERES onto the vacuole.

Atg14p's role on the vacuole surface is likely related to the formation of large $L_o$ domains (see *Figure 9*), which are proposed to function as entry portals for LD uptake into the vacuole (*Wang et al., 2014*). We observed that LDs dock exclusively at $L_o$ domains on the vacuole surface. That Atg14p helps stabilize/form these large $L_o$ domains is supported by our findings that: (1) Atg14p localizes on the rims of these domains, (2) deletion of Atg14p abolishes the domains, and (3) conditions that prevent Atg14p from redistributing onto the vacuole (including deletion of *SNF1*, rapamycin treatment, or gradual GR) diminishes the $L_o$ domains and results in no LD uptake into the vacuole.

In examining the subcellular distribution of other autophagy-related PI3K complex I components (including Atg6p, Atg38p, Vps15p, Vps34p and Vps38p), we found that only Atg6p resides on the vacuole surface with Atg14p under acute GR conditions. Atg6p interacts with Atg14p to form a PI3K complex involved in autophagy (*Kihara et al., 2001*), so we studied Atg6p's distribution under different starvation conditions. Unlike Atg14p, Atg6p resided on the vacuole whether cells were fed or starved. Under acute GR, when large $L_o$ domains formed, Atg6p localized within these large domains. Deletion of *ATG6* prevented any $L_o$ domain formation under acute GR. This suggested Atg6p works in conjunction with Atg14p to generate the large $L_o$ domains, with Atg14p crucial for triggering $L_o$ domain formation (*Figure 9*). Exactly how Atg14p could stimulate vacuolar micro-domain formation is unclear, but it may involve LDs themselves, which by contributing sterols to the vacuolar surface could stimulate micro-domain formation, as suggested by *Wang et al. (2014)*.

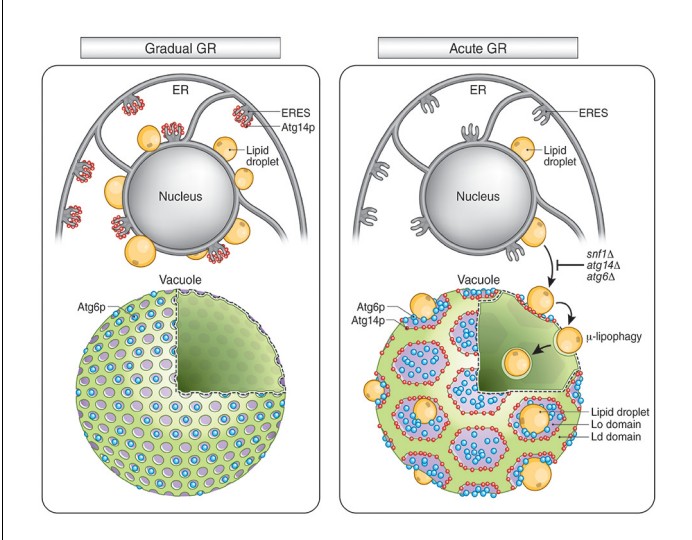

**Figure 9.** Model for AMPK-Atg14p-dependent $L_o$ domain formation on the vacuole and its role in μ-lipophagy during acute GR. In cells undergoing acute GR, LDs and the vacuole coordinate to mediate μ-lipophagy. This response involves AMPK/Snf1p activation and Atg14p redistribution to liquid-ordered membrane ($L_o$) domains on the vacuole surface. Atg14p in cells starved by gradual GR, where AMPK is inactive, does not associate with the vacuole, and instead resides at ER exit sites (ERES). Under acute GR, vacuole-associated Atg14p preferentially resides on the edges of $L_o$ domains and drives large-scale $L_o$ domain formation in concert with Atg6p. The enlarged $L_o$ domains now serve as LD recruitment sites, with LDs moving to the vacuole from the perinuclear ER area. LDs on the $L_o$ domain then bud into the vacuolar lumen and undergo degradation to recycle stored fat molecules, resulting in the extension of lifespan of starved cells during acute GR.

LDs play other role(s) during starvation than simply providing an alternate carbon source for energy production when glucose is absent. This includes supplying lipids for bulk autophagy (*Dupont et al., 2014*; *Shpilka et al., 2015*), and serving as a reservoir for stress-induced excess/ damaged lipids (*Velázquez et al., 2016*). While important, these other LD functions are not sufficient for cells to survive long-term under nutrient starvation unless LD consumption is occurring by μ-lipophagy. Evidence supporting this came from our findings that in glucose-starved cells in which μ-lipophagy is absent (i.e. *atg14Δ* cells, *snf1Δ* cells, cells under gradual GR, or rapamycin-treated cells), less energy is produced relative to cells engaged in μ-lipophagy, and cells survive only short-term. Selective autophagy pathways such as mitophagy or pexophagy have been suggested to be important in the long-term survival of starved cells through organellar quality control (*Kanki et al., 2011*; *Seo et al., 2010*; *Suzuki, 2013*). That μ-lipophagy is more significant than either of these for cell longevity under starvation by acute GR conditions was suggested by our findings that in mitophagy-deficient (e.g. *atg32Δ* and *atg33Δ*) cells or pexophagy-deficient *atg11Δ* cells, μ-lipophagy and long-term survival still occurred.

In summary, our data demonstrate that AMPK and vacuole-associated Atg14p activities play critical roles in cell survival under acute GR by triggering μ-lipophagy (*Figure 9*). The key step for this process is early onset AMPK signaling caused by acute GR. This leads to the relocation of Atg14p, which also physically interacts with AMPK/Snf1p, from ERES to the vacuole. On the vacuole surface, Atg14p initiates μ-lipophagy by working in conjunction with Atg6p and other core autophagy machinery. Through their cooperative roles in promoting μ-lipophagy, these components enable yeast cells undergoing acute GR to dramatically extend their lifespans relative to cells experiencing gradual GR or rapamycin treatment. Cells only preserved Snf1p activity and stimulated μ-lipophagy when Snf1p was activated as a consequence of acute glucose depletion (no similar response occurred when TOR was deactivated by amino acid loss or rapamycin treatment). This supports the notion that AMPK activation selectively promotes autophagic recycling of stored carbon-energy sources (e.g. lipids), while TOR inhibition activates autophagy pathways leading to recycling of nitrogen sources (i.e. digestion of protein-enriched compartments). Thus, under different nutrient stresses, yeast cells choose distinct autophagic pathways for digestion of different substrates (e.g. lipids versus amino acids) to optimize cell metabolism and survival. Given the shared cellular machinery between yeast and other organisms, these findings are germane to understanding cell survival mechanisms under different starvation regimens and should be relevant for clarifying cellular mechanisms for abnormal metabolic processes associated with cancer, obesity and aging.

## Materials and methods

### Yeast strains, plasmids, and media

Yeast BY strains and plasmids used in this study are listed in *Tables 1* and *2*. Rich medium (YPD) containing 2% glucose and synthetic dextrose minimal (SD) or complete (SC) media containing either 2% glucose or 0.4% glucose were prepared as described by *Alvers et al. (2009a)*. YPD medium containing 1% (wt/vol) Bacto yeast extract, 2% (wt/vol) Bacto peptone, and 2% (wt/vol) dextrose was prepared. Synthetic base medium containing 0.17% (wt/vol) Difco yeast nitrogen base without amino acids and ammonium sulfate, 0.5% (wt/vol) ammonium sulfate, and 10 mM $K_2HPO_4$ was prepared at 80% of the final volume. Solid plates were made by adding 1.6% (wt/vol) Bacto agar. For synthetic complete media, 0.067% (wt/vol) of Complete Supplement Mixture minus leucine and uracil (MP Biomedicals) was prepared using the aforementioned synthetic base medium. These media were autoclaved at 121°C for 15 min. Stock solutions for supplemental amino acids, nucleic acid bases, and dextrose (20% wt/vol) were prepared according to Sherman et al. (*Sherman, 2002*), were filter-sterilized, and were added as needed to make SD or SC media. A final concentration (250 μg/mL) of the antibiotic $G_{418}$ sulfate (Mediatech Inc.) was supplied if possible to culture media except the culture conditions for CWY7183, CWY7226, AYS1701, AYS1703, AYS1704, AYS1705, and AYS1706. The final volume was adjusted to 100% with sterile water. To generate pUC19-URA3-3'SEC13-yemRFP, yeast-enhanced mRFP (yemRFP) was amplified from pRS415-yemRFP by PCR using the following primers: 5'-aatgggaacccgctggtgaagttcatcagtacgatccaccggtcgccaccatggtttcaaaaggtgaagaagataatatggc and 3'-ttttcttttgagatgtttcattttaaattcttgatactacggccgctttatttatataattcatccataccaccagttgaatgtc. The PCR product was used to replace GFP from pUC19-URA3-3'SEC13-GFP (*Rossanese et al., 1999*).

**Table 1.** List of yeast strains.

| Strain | Genotype | Source |
|---|---|---|
| BY4742 | MATα, his3Δ1, leu2Δ0, lys2Δ0, ura3Δ0 | (Brachmann et al., 1998) |
| WT | BY4742 hoΔ::KanMX4 | (Winzeler et al., 1999) |
| snf1Δ | BY4742 snf1Δ::KanMX4 | EUROSCARF |
| snf4Δ | BY4742 snf4Δ::KanMX4 | " |
| atg1Δ | BY4742 atg1Δ::KanMX4 | " |
| atg2Δ | BY4742 atg2Δ::KanMX4 | " |
| atg3Δ | BY4742 atg3Δ::KanMX4 | " |
| atg5Δ | BY4742 atg5Δ::KanMX4 | " |
| atg6Δ | BY4742 atg6Δ::KanMX4 | " |
| atg7Δ | BY4742 atg7Δ::KanMX4 | " |
| atg8Δ | BY4742 atg8Δ::KanMX4 | " |
| atg9Δ | BY4742 atg9Δ::KanMX4 | " |
| atg10Δ | BY4742 atg10Δ::KanMX4 | " |
| atg11Δ | BY4742 atg11Δ::KanMX4 | " |
| atg12Δ | BY4742 atg12Δ::KanMX4 | " |
| atg13Δ | BY4742 atg13Δ::KanMX4 | " |
| atg14Δ | BY4742 atg14Δ::KanMX4 | " |
| atg15Δ | BY4742 atg15Δ::KanMX4 | " |
| atg16Δ | BY4742 atg16Δ::KanMX4 | " |
| atg17Δ | BY4742 atg17Δ::KanMX4 | " |
| atg18Δ | BY4742 atg18Δ::KanMX4 | " |
| atg19Δ | BY4742 atg19Δ::KanMX4 | " |
| atg20Δ | BY4742 atg20Δ::KanMX4 | " |
| atg21Δ | BY4742 atg21Δ::KanMX4 | " |
| atg22Δ | BY4742 atg22Δ::KanMX4 | " |
| atg23Δ | BY4742 atg23Δ::KanMX4 | " |
| atg24Δ | BY4742 atg24Δ::KanMX4 | " |
| atg26Δ | BY4742 atg26Δ::KanMX4 | " |
| atg27Δ | BY4742 atg27Δ::KanMX4 | " |
| atg29Δ | BY4742 atg29Δ::KanMX4 | " |
| atg31Δ | BY4742 atg31Δ::KanMX4 | " |
| atg32Δ | BY4742 atg32Δ::KanMX4 | " |
| atg33Δ | BY4742 atg33Δ::KanMX4 | " |
| atg34Δ | BY4742 atg34Δ::KanMX4 | " |
| atg36Δ | BY4742 atg36Δ::KanMX4 | " |
| tgl3Δ | BY4742 tgl3Δ::KanMX4 | " |
| tgl4Δ | BY4742 tgl4Δ::KanMX4 | " |
| AYS1501 | ATG14-3xGFP::HIS WT | This study |
| AYS1503 | ATG14-3xGFP::HIS snf1Δ | " |
| AYS1506 | ATG6-3xGFP::HIS WT | " |
| AYS1508 | ATG6-3xGFP::HIS snf1Δ | " |
| AYS1601 | ATG14-3xGFP::HIS SEC13-yemRFP::URA3 WT | " |
| AYS1602 | ATG14-3xGFP::HIS SEC13-yemRFP::URA3 snf1Δ | " |
| CWY7183 | ATG6-3xGFP::HIS VPH1-mCherry:LEU BY4742 | (Wang et al., 2014) |

*Table 1 continued on next page*

*Table 1 continued*

| Strain | Genotype | Source |
|--------|----------|--------|
| CWY7226 | *ATG14-3xGFP::HIS VPH1-mCherry:LEU* BY4742 | " |
| YPJ1075 | BY4742 *dga1Δ lro1Δ::KanMX4* | (*Petschnigg et al., 2009*) |
| YPJ1076 | BY4742 *are1Δ are2Δ::KanMX4* | " |
| YPJ1078 | BY4742 *dga1Δ lro1Δ are1Δ are2Δ::KanMX4* | " |
| BY4741 | *MATa, his3Δ1, leu2Δ0, met15Δ0, ura3Δ0* | (*Brachmann et al., 1998*) |
| AYS1701 | *ATG6-3xGFP::HIS p416-Ivy1-mCherry* WT | This study |
| AYS1703 | BY4741 *ATG38-GFP::HIS p416-Ivy1-mCherry* | " |
| AYS1704 | BY4741 *VPS15-GFP::HIS p416-Ivy1-mCherry* | " |
| AYS1705 | BY4741 *VPS34-GFP::HIS p416-Ivy1-mCherry* | " |
| AYS1706 | BY4741 *VPS38-GFP::HIS p416-Ivy1-mCherry* | " |

EUROSCARF, European *Saccharomyces cerevisiae* Archive for Functional Analysis, Institute for Molecular Biosciences, Johann Wolfgang Goethe-University Frankfurt, Frankfurt, Germany.

## Yeast growth, glucose restriction, and genetic manipulations

Glucose restriction (GR) experiment was performed with SD and SC media containing either 2% (wt/vol) glucose (gradual GR) or 0.4% (wt/vol) glucose (acute GR). For non-starvation (no GR) conditions, yeast cells grown under YPD or SD and SC media containing 2% glucose were analyzed when they are in an early log phase ($OD_{600} < 0.4$). Briefly, yeast strains from frozen permanent stocks at $-80°C$ were patched onto YPD agar plates. After 2 days of growth at 30°C, cells from patches were inoculated into 5 mL of YPD in 14 mL polypropylene tubes and grown overnight at 30°C in a shaker at ~250 rpm. After ~18 hr, the yeast culture was diluted 1/100 into 5 mL of 2% glucose SD (or SC) medium and grown overnight at 30°C in a shaker at ~250 rpm. After ~18 hr, 0.05 ~ 0.1 OD Unit ($OD_{600} \times$ mL) of yeast cells was transferred to fresh 5 mL of either 2% glucose SD (or SC) for gradual GR or 0.4% glucose SD (or SC) for acute GR. Then, cells were grown at 30°C in a shaker at ~250 rpm without nutrient replenishment to induce glucose starvation. In the case of LD defective cells (i.e. LD*Δ*, SE*Δ*, and TAG*Δ* cells), which completely failed to grow under the above gradual or acute GR conditions, glucose starvation was induced by transferring cells grown under YPD media directly to 2% glucose SD. The $OD_{600}$ was routinely measured on days 1, 2, and 3 to confirm culture saturation

**Table 2.** List of plasmids.

| Plasmid | Organelles | Reference |
|---------|------------|-----------|
| pCuGFP-AUT7 | Autophagosome | (*Kim et al., 2001*) |
| pRS316-PGK-POT-GFP | Peroxisome | (*Binns et al., 2006*) |
| pERG6-mDsRed | Lipid droplet | " |
| pRS416 Vph1-GFP | Vacuole | (*Dawaliby and Mayer, 2010*) |
| pRS426 GFP-Pho8 | Vacuole | " |
| p416-Ivy1-mCherry | Vacuole | (*Toulmay and Prinz, 2013*) |
| pVT100U-mtGFP | Mitochondria | (*Westermann and Neupert, 2000*) |
| pBS-ATG6-3xGFP-His3 | Autophagosome | (*Wang et al., 2014*) |
| pBS-ATG14-3xGFP-His3 | Autophagosome | " |
| pRS416-$P_{GPD}$-GFP-Pho8Δ60-$T_{cyc1}$ | | " |
| pRS315-SNF1-3xHA | | (*McCartney and Schmidt, 2001*) |
| pUC19-URA3-3'SEC13-yemRFP | ER exit site | This study |

by using Beckman DU-640 spectrophotometer (Beckman Coulter Inc.). The transformants were generated as described using EZ-Yeast Transformation Kit (MP Biomedicals).

## Drug treatment

Rapamycin (Cell Signaling Technology Inc.) was dissolved in dimethyl sulfoxide (DMSO) to make 1 mM stock solution and added to SC media containing 2% or 0.4% glucose at 40, 50, 100, 250, or 500 nM concentration. MG132 (Sigma-Aldrich Inc.) was dissolved in DMSO to make 10 mM stock solution and added to SD medium with 2% glucose at 10 µM concentration. For vacuole staining, CMAC-Ala-Pro (7-amino-4-chloromethylcoumarin-L-alanyl-L-proline amide, Thermo Scientific Inc.) was dissolved in DMSO to make 10 mM stock solution and added to 200 µL of yeast cultures at 100 µM concentration, followed by 4 hr incubation at 30°C before imaging experiments. For lipid droplet staining, Nile red (Thermo Scientific Inc.) was dissolved in DMSO to make 1 mg/mL stock solution and added to 500 µL of yeast cultures at 1 µg/mL concentration, followed by 15 min incubation at 30°C before imaging experiments.

## Cell lifespan, cellular respiration and stress resistance measurements

Cell lifespan/viability was evaluated by using a vital dye (i.e. erythrocin B) or by the method of *Alvers et al. (2009b)*. The rate of oxygen consumption was monitored using a Clark-type oxygen electrode (Oroboros Instruments Corp.). During a cell lifespan experiment, cell cultures (1.5 mL) were transferred to an airtight chamber maintained at 30°C with magnetic stirrer, and oxygen content was monitored for at least 20 min according to the manufacturer's instructions with slight modification. The rate of oxygen consumption was normalized with $OD_{600}$. Sensitivity to oxidative stress-inducing agents (i.e. 2.5 mM hydrogen peroxide and 30 µM menadione) was measured by determining colony forming units (CFUs) per mL of yeast culture. Briefly, 1 hr prior to stress resistance test, YPD agar plates with or without previously indicated oxidative stressors were freshly prepared. During lifespan measurement, a serial fivefold diluted culture was plated onto the aforementioned agar plates. These replica plates were incubated further at 30°C for 3 days to visualize viable colonies.

## Autophagic flux and LD degradation assays

Degradation flux of autophagosomes, cytosolic autophagic substrates, peroxisomes, and LDs were performed according to Alvers et al. (*Alvers et al., 2009b*) with slight modification using α-RFP (Ajay Sharma, CBMP, NICHD) or α-GFP (Abcam, # ab290) antibodies.

## Yeast cell lysis, immunoprecipitation, AMPK phosphorylation analysis

Cell lysates were prepared by a glass-bead method using immunoprecipitation (IP) buffer (0.1% NP-40 (4-nonylphenyl poly(ethyleneglycol), 50 mM Tris-HCl pH8, 150 mM NaCl, and 0.1 mM DTT (4-dithiothreitol)) or CHAPS buffer (1% CHAPS (3-[(3-cholamidopropyl) dimethylammonio]-1-propane-sulfonate), 25 mM Hepes, 2 mM $MgCl_2$, and 100 mM NaCl) with Halt Protease and Phosphatase Inhibitor Cocktail (Thermo Scientific Inc.). For IP experiments, 25 OD Unit of yeast cells were collected and washed once with cold IP or CHAPS buffers. Cell pellets were resuspended with 0.5 mL IP or CHAPS buffers in 2-mL microfuge tube containing 0.25 g of 0.5 mm glass beads (Sigma-Aldrich Inc.). Then, the tube was placed on a vortex mixer at 4°C for 20 min. Cell lysates were centrifuged at top speed at 4°C for 15 min and supernatants were transferred to fresh tube with 30 µL of Protein A-Sepharose4B (Thermo Scientific Inc.) pre-treated with 2 µL of rabbit α-GFP (Ajay Sharma, CBMP, NICHD) or total rabbit IgG (Thermo Scientific Inc.). All tubes were incubated at 4°C on a rotator. After 1 hr incubation, beads were pelleted at 10,000 rpm for 10 s at 4°C and then resuspended and washed 3 times in 1 mL cold IP or CHAPS buffers. To remove NP-40, beads were washed twice in IP buffer without NP-40 and DTT. After removing all remaining buffer, beads were resuspended with 60 µL of 2X Laemmli sample buffer, boiled for 5 min, and centrifuged at top speed at 4°C for 5 min. Samples were stored at −80°C prior to performing Western blot analysis.

For AMPK phosphorylation experiments, 5 OD Unit of yeast cells were centrifuged at 2000 rpm at 4°C for 5 min and the pellets were immediately frozen by liquid nitrogen. Using cold IP buffer containing 0.25 g of 0.5 mm glass beads, cell pellets were homogenized to extract proteins as described in the above. Afterwards, BCA protein assay (Thermo Scientific Inc.) was performed to determine protein concentrations. The same amount of proteins (i.e. 10 or 15 µg) was used for

western blot analysis. In addition, protein loading was monitored using TGX Stain-Free Precast Gels System (Bio-Rad Laboratories) according to the manufacturer's instructions. Phosphorylated and total Snf1p was determined as modified from McCartney and Schmidt (*McCartney and Schmidt, 2001*) using α-phospho AMPK (Cell Signaling, #2535) and α-HA tag HRP (Abcam, #ab128131) antibodies, respectively.

## Cell survival measurements under rapamycin

Freshly prepared yeast patch was inoculated into 5 mL of SC media containing 2% glucose and was grown at 30°C in a shaker at ~250 rpm. After ~18 hr, the cell culture was diluted 1/100 into 5 mL of SC media containing 2% or 0.4% glucose. After ~18 hr, cells were diluted by the equal volume of the corresponding media that contain 0.5 or 1 µM rapamycin, followed by growing them at 30°C in a shaker at ~250 rpm. At the indicated times, 0.00046 OD Unit cells were plated onto YPEG (1% Bacto yeast extract, 2% Bacto peptone, 2%(vol/vol) ethanol, 3%(wt/vol) glycerol) agar plate after four serial fivefold dilutions. These plates were incubated further at 30°C for 4 days to visualize viable colonies.

## ATP measurements

Two OD Unit of yeast cells are collected in 1.5-mL microfuge tube by centrifugation at top speed for 1 min and washed once with cold distilled water. Supernatant was removed and 0.75 mL of 90% acetone was added to the pellet. Resuspended cells were heated at 90°C to allow the acetone to evaporate off for 30 min. Approximately 50 µL of solution is remaining. Assay buffer (10 mM Tris pH8.0 and 1 mM EDTA) was added to make it to the final volume of 500 µL. The concentration of ATP was then measured according to ATP Determination Kit (Thermo Scientific Inc.) by following the manufacturer's instruction using a luminescence microplate reader (BioTek U.S.).

## Microscopy

Soft X-ray images of cryogenic yeast samples were collected using XM-2, the National Center for X-ray Tomography soft X-ray microscope at the Advanced Light Source of Lawrence Berkeley National Laboratory and reconstructed according to previously published protocols (*Larabell and Nugent, 2010*). Organelles were segmented for data analysis according to *Uchida et al. (2011)*. Epifluorescence images of live cells were acquired using EVOS FL Cell Imaging System (Thermo Scientific Inc.) equipped with DAPI (357/44 Ex; 447/60 Em), GFP (470/22 Ex; 510/42 Em), and RFP (531/40 Ex; 593/40 Em) light cubes. Confocal fluorescence images of live cells were acquired using Leica SP5 equipped with white light laser (WLL) (Leica Microsystems) or a customized Nikon TiE inverted scope equipped with a Yokogawa spinning-disk head (#CSU-X1, Yokogawa) and a Photometrics EM-CCD camera (Evolve 512). 488 and 561 nm or 563 nm WLL lines were used for exciting GFP and DsRed, respectively, and the emission signals were sequentially collected to minimize crosstalk. To image cells, 5 µL of cell cultures was mounted directly under a coverglass or 200 µL of cell cultures were plated and incubated in NuncLab-Tek II Chambered Coverglass (Thermo Scientific Inc.) at RT. Fixed cells were imaged using Zeiss LSM 880 Airyscan microscopy with a 63× Plan-Apochromat 1.4 NA oil objective (Carl Zeiss). DAPI, GFP, and DsRed were excited with 405, 488, and 561 nm lines, respectively, and the raw image data was processed using the commercial Zen software package (Carl Zeiss) and Imaris software (Bitplan), or ImageJ (National institutes of Health).

## Yeast fixation for AIRY-scan imaging

Cells were fixed using cold 4% paraformaldehyde with 3.4% sucrose at RT for 15 min. Then, cells were washed twice with KPO4/Sorbitol (0.1M potassium phosphate pH7.5 1.2M sorbitol). Afterwards, cell pellets were resuspended in 50 µL KPO4/Sorbitol with DAPI at 1 µg/mL concentration and were incubated at RT for 10 min. After washing twice with KPO4/Sorbitol, cells were resuspended in Vectashield Mounting Medium (Vector Laboratories), were mounted directly under a coverglass, and were placed at RT for >3 hr to stabilize the mount before performing imaging experiments.

## Statistical analysis

Statistical analysis was performed using Prism four for Student's t-test and Grubbs' test (GraphPad Software Inc.). The significance level was set at $p < 0.05$.

## Acknowledgements

We thank to the members of the Lippincott-Schwartz lab for helpful discussions.

## Additional information

### Funding

| Funder | Grant reference number | Author |
|---|---|---|
| Howard Hughes Medical Institute | | Jennifer Lippincott-Schwartz |
| National Institutes of Health | | Jennifer Lippincott-Schwartz |
| National Institute of General Medical Sciences | P41GM103445 | Carolyn A Larabell |
| U.S. Department of Energy | DE-AC02-5CH11231 | Carolyn A Larabell |

The funders had no role in study design, data collection and interpretation, or the decision to submit the work for publication.

### Author contributions

AYS, Conceptualization, Resources, Data curation, Formal analysis, Supervision, Validation, Investigation, Visualization, Methodology, Writing—original draft, Project administration, Writing—review and editing; P-WL, DF, PS, Formal analysis; MALG, BC, CAL, Methodology; JL-S, Conceptualization, Resources, Supervision, Funding acquisition, Writing—original draft, Project administration, Writing—review and editing

### Author ORCIDs

Arnold Y Seo, http://orcid.org/0000-0002-4607-8925
Carolyn A Larabell, http://orcid.org/0000-0002-6262-4789
Jennifer Lippincott-Schwartz, http://orcid.org/0000-0002-8601-3501

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
