## [Decision Letter]

Thank you for submitting your article "Atg14 Redistributes from ER Exit Sites to Vacuolar Domains to Drive Lipophagy upon AMPK Activation by Glucose Starvation" for consideration by *eLife*. Your article has been reviewed by 2 peer reviewers, and the evaluation has been overseen by Vojo Deretic as the Reviewing Editor and Vivek Malhotra as the Senior Editor. The reviewers have opted to remain anonymous.

The reviewers have discussed the reviews with one another and the Reviewing Editor has drafted this decision to help you prepare a revised submission.

Summary:

In this work, the authors describe how yeast carry out, under certain physiological conditions, a variant of lipophagy (autophagic degradation of neutral lipid stores often referred to as lipid droplets) through vacuolar uptake of lipid droplets/bodies. This process morphologically resembles microautophagy and in its genetic requirements and requires Atg14 re-positioning from ER exit sites to the vacuole. Curiously, inhibition of mTOR inhibits this process (unlike in the majority of other autophagic processes where mTOR conventionally plays a negative regulatory role).

The initial reviews ranged from significant to minimal experiments and revisions. This led to a discussion that resulted in the following consensus recommendation:

Please address points 1 and 2 experimentally (with details clearly outlined by the reviewers in the section reporting complete reviewers' comments), and point 3 textually (details from reviewing editor given below).

Essential revisions:

1) The events post Atg14 association to the vacuole in the clearance of lipid droplet are not clear. The data that Atg14 localizes to ergosterol enriched domains is not convincing (Figure 7). What other components are recruited to the ergestorol enriched vacuolar domains? The details are clearly outlined by the reviewers in the section reporting complete reviewers' criticisms.

2) Is the effect of lipophagy on cell survival direct? The reviewers have suggested experiments that could potentially help address this concern. What is the effect of deletion of regulators of Snf1 on cell survival? The details are clearly outlined by the reviewers in the section reporting complete reviewers' criticisms.

Complete reviewer comments:

*Reviewer #1:*

In the present manuscript, Seo et al. delineate principle regulatory differences of yeast cells transitioning to stationary phase in the presence of high (2%) or medium (0.4%) glucose concentrations critically determining chronological lifespan. Specifically, yeast cells activate a sustained autophagy response, which includes the selective turnover of lipid droplets (lipophagy), when AMP-dependent kinase Snf1 is stabilized and activated in the presence of low glucose (0.4%). Snf1 activity is required for re-localization of Atg14, a subunit of the phosphatidylinositol-3 kinase complex I that can also physically interact with Snf1 as shown in this manuscript, from ER exit sites to vacuoles and function in lipophagy without affecting general autophagy. Snf1 and Atg14 function is required for the long lifespan of yeast cells during chronological aging in the presence of low glucose concentrations.

The manuscript presents data that indicate important differences in the regulation of autophagy and cell metabolism dependent on glucose concentrations and critical consequences for cell survival during starvation. Furthermore, the data further evidence for fundamental plasticity in the regulation and function of core autophagy components defined by metabolic or stress conditions. Hence the findings in the present study will be of great interest in the autophagy and aging field. However, before supporting publication, two main points need to be addressed.

1) What is the relative contribution of sustained general autophagy and selective lipophagy for cell survival?

The authors provide evidence indicating that conditions inducing lipophagy are correlated with longevity during acute GR. This conclusion depends mainly on the phenotypes associated with deletion of Snf1 or Atg14. However, it is important to note that the deletion of atg14 strongly compromises non-selective turnover of a cytosolic model substrate (GFP-Pho8d60) after two days of acute GR raising the possibility that general autophagy defects contribute to or cause shortened lifespan of this mutant. In addition, AMPK/Snf1 deletion compromises the regulation of a number of cellular functions other than lipophagy including fatty acid synthesis, mitochondrial activity, and b-oxidation of fatty acids, which likely contribute to the shortened phenotype observed for this mutant.

First, to strengthen the postulated role for lipophagy in lifespan regulation, the authors should measure wild type cell survival in SC media in the presence or absence of rapamycin during gradual and acute GR. As shown in Figure 3, addition of rapamycin does not affect the general autophagy response judged by GFP-Atg8 turnover, but selectively reduces lipophagy (Erg6-DsRed turnover) to rates observed for gradual GR. Thus, if lipophagy plays a central role under these conditions, one would expect a dramatically shortened lifespan upon addition of rapamycin during acute GR.

Second, the authors show that deletion of atg14 decreases cell survival during acute GR, but do not formally demonstrate that defects in other core autophagy components (*atg1, atg5, atg23*) reduce lifespan.

Third, the authors should test whether lipid droplets are required for long chronological lifespan by testing cells that are defective in the biogenesis of lipid droplets.

2) The lipophagy-specific function of Atg14 is unexpected and intriguing. However, in which capacity Atg14 functions in lipophagy remains underdeveloped, since the authors included only Atg6 in their analysis. The authors should analyse the remaining components of the PI3 kinase complexes including Vps34, Vps15, Vps38, and Atg38.

The physical interaction between Atg14 and Snf1 is interesting, but its significance is unclear. Is this interaction specific for acute GR or does it occur under all conditions? Are Snf1 and other core autophagy components recruited to the vacuole in an Atg14 dependent manner?

The authors convincingly show that Snf1 activity is required for lipophagy and longevity during acute GR, but is it sufficient? The authors should test whether yeast mutants that show constitutive Snf1 activity (deletion of *reg1* or equivalent strains) render cells long lived during gradual GR.

The authors need to include a wild type control Figure 6.

*Reviewer #2:*

The authors investigate the mechanisms underlying lipid metabolism during starvation in budding yeast. Specifically, they examine the effects of acute versus gradual glucose restriction and describe differences between these two starvation conditions. Amongst the differences, they focus on lipophagy, which specifically occurs under acute glucose starvation conditions. Their characterizations suggest that AMPK is the selective upstream regulator of Atg14-dependent lipophagy, as opposed to bulk autophagy, under acute glucose restriction. More preliminary localization data suggest that Atg14 is recruited to the vacuole at ergosterol rich domains in an AMPK dependent manner where it functions to recruit lipid droplets for turnover.

The results shown in Figure 1 are descriptive and are not integrated with the rest of the manuscript.

Is gradual glucose starvation the same as growing the cells to stationary phase?

The observation that rapamycin treatment blocks lipophagy during glucose starvation is contrary to what has been previously observed. It has been reported that TORC1 is inhibited when cells are starved for glucose based on assays using Sch9 phosphorylation. In addition, *snf1* deletion slows, but doesn't block the decrease in TORC1-dependent Sch9 phosphorylation in response to glucose starvation (Figure 5http://www.genetics.org/content/198/2/773). The work would benefit from a more detailed characterization of the apparent antagonizing roles of AMPK and TORC1 in regulating differences between nutrient sensing (amino acids vs carbon source starvation) and types of autophagy.

What is the mechanism for Atg14-dependent lipophagy? The authors state "Once vacuole-associated, Atg14p and other autophagy factors (including Atg6p) create sites on the vacuole surface where LDs can bind and internalize by an autophagic process." What exactly is this process? The authors speculate that is a form of microautophagy. In this context, the conclusion that Atg14 localizes to egosterol enriched vacuolar domains is not sufficiently supported by data presented in Figure 7. Without a functional context, these data are too preliminary. The manuscript would be improved by focusing on the mechanism of lipophagy.

---

## [Author Response]

*Essential revisions:*

*1) The events post Atg14 association to the vacuole in the clearance of lipid droplet are not clear. The data that Atg14 localizes to ergosterol enriched domains is not convincing (Figure 7). What other components are recruited to the ergestorol enriched vacuolar domains? The details are clearly outlined by the reviewers in the section reporting complete reviewers' criticisms.*

To further characterize the events post-Atg14p association to the vacuole, we a) provide more convincing data that Atg14p localizes to ergosterol-enriched domains (which we now refer to as L_o_ vacuole domains), b) demonstrate what other components are recruited to these L_o_ domains, and c) address other concerns related to this topic raised by the reviewers.

A) To provide more convincing data that Atg14 localizes to L_o_ domains, we imaged Atg14-GFP expressing cells undergoing acute GR by AIRY-scan superresolution microscopy and analyzed the data using a 3D volume rendering method (Imaris). With the improved spatial resolution and 3D renderings of Atg14p and Vph1p (or Pho8p) (i.e., liquid-disordered membrane (L_d_) domain markers (Toulmay and Prinz, 2013)), we confirmed that Atg14p localizes to L_o_ domains on the vacuolar membrane. We further showed that LDs, labeled using Erg6-DsRed, localize exclusively in large L_o_ vacuole membrane domains prior to being internalized into the vacuole in cells undergoing acute GR (Figure 8). This suggests L_o_ domains serve as preferential sites for LD uptake into the vacuole. By contrast, in cells undergoing gradual GR, in which Atg14p localizes to ERES instead of the vacuole membrane, L_o_ domains on the vacuole were much less apparent and LDs remained cytoplasmic, with none taken up into the vacuole (Figure 8).

We also used AIRY-scan confocal microscopy to examine the vacuolar localizations of Atg14p, Atg6p and a vacuole L_d_ domain marker under acute GR. 3D volume reconstruction and maximum projection images revealed Atg14p molecules localizing as punctate elements within vacuolar L_o_ domains (Video 3 and Figure 8). Single z-slices from these images showed Atg14p was preferentially localized on the edges of L_o_ domains (Figure 8, Single z-slice), in close association with LDs (Figure 8—figure supplement 1). Atg6p was also localized in L_o_ domains at the vacuole surface but was more broadly distributed relative to Atg14p (Figure 8).

To address whether Atg14p and/or Atg6p are critical for creating the large L_o_ domains on vacuole membranes under acute GR, we deleted the gene for each protein and then examined whether or not L_o_ domains were formed. Little or no L_o_ domains were observed in either *atg14△* or *atg6△* cells under acute or gradual GR, with the effect most potent in *atg6△* cells (Figure 8). Given Atg6p resides on the vacuole membrane irrespective of the starvation condition (see Figure 7), whereas Atg14p redistributes to the vacuole membrane in response to acute GR (Figure 6), the results suggested Atg14p helps drive large-scale L_o_ formation under acute GR, with a critical supportive role played by Atg6p. Consistent with this, fewer large-scale L_o_ domains formed in *snf1△* cells under acute GR (Figure 8), a condition where Atg6p resides on the vacuole surface but Atg14p fails to be recruited (see Figure 6).

B) To demonstrate what other components are recruited to the vacuole under acute GR, we monitored the distribution of GFP-tagged Atg6p, Atg38p, Vps15p, Vps34p, and Vps38p, which are autophagy-related, PI3K complex I components. Vacuole association of each PI3K complex I component was scored by AIRY-scan confocal microscopy image analysis. Prior to glucose starvation, Atg6p and Vps15p both showed significant association with the vacuole surface. Atg6p remained localized to the vacuole after shifting to either acute or gradual GR, but Vps15p redistributed off the vacuole upon acute GR (Figure 7). The other PI3K complex I components (i.e., Atg38p, Vps34p, and Vps38p) remained primarily cytosolic with no increased vacuolar association upon shift to either acute or gradual GR (Figure 7). Therefore, only Atg6p and Atg14p reside on the vacuole surface under acute GR. Atg6p remained prominently localized at the vacuolar surface even when AMPK was inactive (i.e., in *snf1Δ* cells or during gradual GR) (Figure 7), whereas Atg14p was primarily ERES-associated under these conditions (Figure 6). Therefore, only Atg14p’s vacuolar localization was AMPK-dependent.

*2) Is the effect of lipophagy on cell survival direct? The reviewers have suggested experiments that could potentially help address this concern. What is the effect of deletion of regulators of Snf1 on cell survival? The details are clearly outlined by the reviewers in the section reporting complete reviewers' criticisms.*

We provide additional data, including experiments suggested by the reviewers, that now strongly suggests that µ-lipophagy has a direct role in cell survival under starvation. First, when we prevent LDs from forming in cells by knocking out key LD biosynthetic enzymes (i.e., LDΔ, SEΔ, and TAGΔ cells), cell viability under starvation drops significantly (Figure 1—figure supplement 2). Since LDs are absent and no µ-lipophagy occurs in these cells, the results support the idea that µ-lipophagy’s downstream effects (i.e., LD consumption to drive OXPHOS) are essential for cell survival under glucose starvation. Second, removal of Atg14p, the protein we propose helps direct µ-lipophagy, prevents µ-lipophagy without blocking general autophagy, and reduces overall cell viability under acute GR (Figure 5). Third, different conditions that prevented Atg14p from relocating from ERES onto the vacuole (where Atg14p functions in µ-lipophagy) caused reduced cell survival under acute GR. These conditions included: knock-out of AMPK (s*nf1△* cells); inactivation of TOR through rapamycin-treatment; and inactivation of TOR through amino acid removal. Fourth, µ-lipophagy and long-term survival still occurred in mitophagy-deficient (e.g., *atg32Δ* and *atg33Δ*) cells or pexophagy-deficient *atg11Δ* cells undergoing acute GR (Figure 5 and Figure 5—figure supplement 3), indicating that µ-lipophagy is more significant than either mitophagy or pexophagy for cell longevity under starvation by acute GR conditions.

*Complete reviewer comments:*

*Reviewer #1:*

*[…] 1) What is the relative contribution of sustained general autophagy and selective lipophagy for cell survival?*

Our data suggest that both general autophagy and µ-lipophagy are critical for long-term survival under nutrient starvation. When either pathway is compromised or inhibited, long-term survival under starvation is lost. For example, we observed µ-lipophagy is absent and general autophagy is present in both *atg14Δ* cells and cells starved by rapamycin-treatment or gradual GR; and these cells failed to live long-term. We further found that blocking autophagy by knocking out genes for core autophagy machinery (e.g., Atg1p and Atg5p) significantly reduced long-term cell viability under acute GR.

As mentioned above, a major point of our study is that cells switch on both µ-lipophagy and bulk autophagy pathways under acute GR to allow selective recycling of LDs (via µ-lipophagy) as an alternative energy source and recycling of proteins (via bulk autophagy) as a nitrogen source. Mitophagy or pexophagy were not needed since blocking either of these pathways by deleting *ATG11* or *ATG32*, respectively, did not decrease long-term cell survival under acute GR (Figure 5 and Figure 5—figure supplement 3).

*The authors provide evidence indicating that conditions inducing lipophagy are correlated with longevity during acute GR. This conclusion depends mainly on the phenotypes associated with deletion of Snf1 or Atg14. However, it is important to note that the deletion of atg14 strongly compromises non-selective turnover of a cytosolic model substrate (GFP-Pho8d60) after two days of acute GR raising the possibility that general autophagy defects contribute to or cause shortened lifespan of this mutant.*

We agree that general autophagy defects may contribute to the shortened lifespan of the *Atg14△*cells, but we believe a major defect is from blockade of µ-lipophagy. This is based on our observation that other conditions in which cells are unable to switch on µ-lipophagy (e.g., rapamycin-treatment or gradual GR) result in cells having shortened lifespans. Without µ-lipophagy-mediated LD catabolism, therefore, general autophagy alone is not sufficient to enable long-term cell survival under prolonged glucose starvation. We speculate that one of µ-lipophagy’s downstream roles is to provide energy for sustaining general autophagy or other selective autophagy pathways under long-term starvation. Further studies are needed to test this possibility.

*In addition, AMPK/Snf1 deletion compromises the regulation of a number of cellular functions other than lipophagy including fatty acid synthesis, mitochondrial activity, and b-oxidation of fatty acids, which likely contribute to the shortened phenotype observed for this mutant.*

We agree that these other processes likely contribute to the shortened phenotype observed for the *snf1△* cells.

*First, to strengthen the postulated role for lipophagy in lifespan regulation, the authors should measure wild type cell survival in SC media in the presence or absence of rapamycin during gradual and acute GR. As shown in Figure 3, addition of rapamycin does not affect the general autophagy response judged by GFP-Atg8 turnover, but selectively reduces lipophagy (Erg6-DsRed turnover) to rates observed for gradual GR. Thus, if lipophagy plays a central role under these conditions, one would expect a dramatically shortened lifespan upon addition of rapamycin during acute GR.*

We performed this experiment and found a dramatically shortened lifespan upon addition of rapamycin to cells undergoing acute GR (see Figure 4 and Figure 4—figure supplement 1). As rapamycin treatment reduces µ-lipophagy, this result supports the thesis that µ-lipophagy plays an important role in the regulation of cell survival during glucose depletion. We further measured the lifespans of cells undergoing gradual or acute GR under different amino acid starvation conditions (i.e., SD versus SC). Similar to rapamycin-treatment, amino acid depletion dramatically shortened the lifespans of these cells (Figure 4—figure supplement 1). These new data are now included in the revised manuscript.

*Second, the authors show that deletion of atg14 decreases cell survival during acute GR, but do not formally demonstrate that defects in other core autophagy components (atg1, atg5, atg23) reduce lifespan.*

To address this comment, we have included new data showing that cells lacking *ATG1, ATG5* or *ATG23* genes all have reduced lifespans under acute GR. This is shown in Figure 5—figure supplement 3.

*Third, the authors should test whether lipid droplets are required for long chronological lifespan by testing cells that are defective in the biogenesis of lipid droplets.*

We have tested this and now show that cells defective in LD biogenesis do not survive long-term during glucose starvation. This is in agreement with prior work of (Velázquez et al., 2016) highlighting the LD’s critical role in cell survival. We have included this data in Figure 1—figure supplement 2.

*2) The lipophagy-specific function of Atg14 is unexpected and intriguing. However, in which capacity Atg14 functions in lipophagy remains underdeveloped, since the authors included only Atg6 in their analysis. The authors should analyse the remaining components of the PI3 kinase complexes including Vps34, Vps15, Vps38, and Atg38.*

We performed these experiments in cells expressing GFP-tagged versions of the remaining PI3 kinase complex components. Vps34p, Vps38p, and Atg38p were distributed in the cytoplasm and Atg6p localized on the vacuole whether or not cells were starved. Vps15p, however, redistributed off the vacuole upon acute GR. These results suggest Atg14p’s regulation of µ-lipophagy on the vacuole surface likely requires only Atg6p among the complex components.

*The physical interaction between Atg14 and Snf1 is interesting, but its significance is unclear. Is this interaction specific for acute GR or does it occur under all conditions?*

We found that the Atg14p-Snf1p interaction was not significantly altered by different GR conditions in our IP experiment. This suggests that Snf1p phosphorylation is not required for Atg14p and Snf1p to interact. It is possible, therefore, that some type of posttranslational modification of Atg14p occurs in response to AMPK activation and that this modification is what determines Atg14p’s involvement in µ-lipophagy. This possibility and others are now being investigated.

*Are Snf1 and other core autophagy components recruited to the vacuole in an Atg14 dependent manner?*

To address this comment, we imaged Snf1p (not shown) and other core autophagy factors (including GFP-tagged Atg6p, Atg38p, Vps15p, Vps34p, and Vps38p) in cells undergoing acute GR, in which Atg14p relocates onto the vacuole. None of these components were recruited onto the vacuole under these conditions. Indeed, Vps15p redistributed off the vacuole, while Atg6p remained vacuole-associated whether or not Atg14p was present there.

*The authors convincingly show that Snf1 activity is required for lipophagy and longevity during acute GR, but is it sufficient? The authors should test whether yeast mutants that show constitutive Snf1 activity (deletion of reg1 or equivalent strains) render cells long lived during gradual GR.*

Previous work has shown that cells lacking*REG1* enhance their respiration by activating yeast AMPK, resulting in increased viability over the course of 3 days (Bonawitz et al., 2007). However, when we conducted our longevity assay under gradual GR conditions in the presence and absence of excessive amino acids (i.e., SC gradual GR and SD gradual GR, respectively), loss of *REG1* did not increase cell longevity in the above conditions. Thus, although constitutive Snf1p activity helps delay loss of cell viability over the short-term, this activity may not be sufficient to drive all necessary long-term survival programs. We have included the above data in Figure 4—figure supplement 2.

*The authors need to include a wild type control Figure 6.*

We have now included the wildtype data in Figure 6—figure supplement 3.

*Reviewer #2:*

*[…] The results shown in Figure 1 are descriptive and are not integrated with the rest of the manuscript.*

In response to this comment, we have moved data examining the effects on stress resistance and mitochondrial morphology under acute versus gradual GR to Figure 1—figure supplement 1. Figure 1 now shows cell viability, respiration levels, and the localization of LDs under acute versus gradual GR. This sets up the physiological basis of our model system and is important for integrating conceptually and experimentally other data in our paper. We’ve added a 3D-rendered segmented organelle image to Figure 1 (bottom panels), as well as two supplementary videos (Video 1 and Video 2), to highlight the different LD-vacuole organizations under gradual versus acute GR conditions. To the best of our knowledge, the SXT data in Figure 1 provides the most detailed information of overall subcellular organization of LDs and the vacuole (Video 1 and Video 2) in the yeast autophagy field.

*Is gradual glucose starvation the same as growing the cells to stationary phase?*

Yes.

*The observation that rapamycin treatment blocks lipophagy during glucose starvation is contrary to what has been previously observed.*

Several studies have reported that TOR inactivation by rapamycin or amino acid starvation triggers lipid droplet biogenesis and does not induce vacuolar LD delivery (Velázquez et al., 2016; Wang et al., 2014). These findings support our findings that TOR inactivation prevents LD degradation. The references that the reviewer refers to are not provided so it is difficult to explain any discrepancy. However, if there are any discrepancies, we are happy to investigate them in future work.

*It has been reported that TORC1 is inhibited when cells are starved for glucose based on assays using Sch9 phosphorylation. In addition, snf1 deletion slows, but doesn't block the decrease in TORC1-dependent Sch9 phosphorylation in response to glucose starvation (Figure 5http://www.genetics.org/content/198/2/773). The work would benefit from a more detailed characterization of the apparent antagonizing roles of AMPK and TORC1 in regulating differences between nutrient sensing (amino acids vs carbon source starvation) and types of autophagy.*

We agree with the reviewer that a more detailed characterization of the antagonizing roles of AMPK and TOR would add to the paper. To address this, we now provide more detailed analysis and discussion on this topic. First, we found that AMPK cannot be activated under starvation conditions when TOR is inhibited beforehand (i.e., rapamycin treatment, gradual GR, or acute GR in rapamycin-treated cells). We speculate that this arises because TOR inhibition induces bulk autophagy and causes a surplus of amino acids, raising ATP levels to antagonize AMPK activity. Second, we demonstrate that rapamycin and gradual GR treatments (which each inhibit TOR) cause proteasomal degradation of AMPK. This relates to the finding that mTOR inhibition activates overall protein degradation by the ubiquitin proteasome system (Zhao et al., 2015). Third, we show that addition of rapamycin to cells undergoing acute GR causes AMPK to be degraded and cell longevity to be shortened. We discuss this from the perspective of the role of µ-lipophagy in long-term survival under starvation, and µ-lipophagy’s requirement for AMPK activation. Indeed, the results help explain why treatments that inhibit AMPK activation, despite triggering TOR inactivation and bulk autophagy pathways, are insufficient for promoting long-term cell survival. Overall, our data suggest that the antagonism between AMPK activity and TOR inhibition allows cells to engage in different types of autophagy to suit their metabolic demands under different starvation conditions. Therefore, cells starved of amino acids prior to glucose removal (i.e., gradual GR or rapamycin-treatment) switch on only bulk autophagy pathways to recycle nitrogen sources. In cells starved acutely of glucose before amino acid withdrawal (i.e., acute GR), by contrast, both µ-lipophagy and general autophagy pathways are switched on. Consequently, these cells are able to survive long-term due to LDs being used as an alternative energy source.

*What is the mechanism for Atg14-dependent lipophagy? The authors state "Once vacuole-associated, Atg14p and other autophagy factors (including Atg6p) create sites on the vacuole surface where LDs can bind and internalize by an autophagic process." What exactly is this process? The authors speculate that is a form of microautophagy. In this context, the conclusion that Atg14 localizes to egosterol enriched vacuolar domains is not sufficiently supported by data presented in Figure 7. Without a functional context, these data are too preliminary. The manuscript would be improved by focusing on the mechanism of lipophagy.*

In the revised paper, we have made considerable effort at characterizing the mechanism(s) underlying µ-lipophagy, including further study of the roles of specific autophagy components, and more detailed imaging of LDs, Atg14p and Atg6p localizations on the vacuole. Prior studies have demonstrated that ergosterol is a key factor required for generating liquid-ordered (L_o_) membrane domains on the vacuole surface (Toulmay and Prinz, 2013), and shown that Atg14p co-localizes with Ivy1p, a L_o_ domain marker on the vacuole (Wang et al., 2014). Using AIRY-scan super-resolution imaging combined with 3D volume rendering, we show more clearly that Atg14p localizes to L_o_ vacuolar domains and is excluded from liquid disordered (L_d_) membrane domains (Figure 8 and Video 3). We find Atg14p localizes on the edges of the L_o_ domains, whereas Atg6p is more broadly redistributed to the L_o_ domain (Figure 8). We further show that L_o_ and L_d_ domains do not form in cells lacking either Atg6p or Atg14p, and that the domains are reduced in cells lacking Snf1p. From this and other data in our paper, we speculate that vacuolar-localized Atg14p helps differentiate L_o_ domains, perhaps by serving to link LDs to the vacuole surface, which would allow sterols from the LD to further drive phase partitioning on the vacuole surface. Once LDs are docked on L_o_ domains, the domains internalize with their associated LDs. These findings open up many new questions about µ-lipophagy that future studies should address.